

# Spatial and temporal heterogeneity of soil respiration in a bare-soil Mediterranean olive grove

Sergio Aranda-Barranco[1,2], Penélope Serrano-Ortiz[1,2], Andrew S. Kowalski[2,3], Enrique P. Sánchez-Cañete [2,3]

[1] Department of Ecology, University of Granada, 18071 Granada, Spain.
[2] Andalusian Institute for Earth System Research (CEAMA-IISTA), University of Granada, 18006 Granada, Spain
[3] Department of Applied Physics, University of Granada, 18071 Granada, Spain

*Correspondence to*: Sergio Aranda-Barranco (sergioaranda@ugr.es)

**Abstract.** Soil respiration ($R_s$) is an important carbon flux in terrestrial ecosystems and knowledge about this $CO_2$ release

process and the drivers involved is a key topic in the context of global change. However, temporal, and spatial variability has not been extensively studied in semiarid systems such as olive groves. In this study, we show a full year of continuous measurements of $R_s$ with six automatic chambers in a fertirrigated olive grove with bare soil in the Mediterranean accompanied by ecosystem respiration ($R_{eco}$) obtained using the eddy covariance (EC) technique. To study spatial variability, the automatic chambers were distributed equally under the canopy ($R_{s\ Under-Tree}$) and in the center of the alley ($R_{s\ Alley}$), and the

gradient of $R_s$ between both locations was measured in several manual campaigns in addition to azimuthal changes about the center of the olive trees. The results indicate that $R_{s\ Under-Tree}$ was three times larger than $R_{s\ Alley}$ in the annual computations. Higher $R_s$ was found on the south face, and an exponential decay of $R_s$ was observed until the alley's center was reached. These spatial changes were used to weigh and project $R_s$ to the ecosystem scale, whose annual balance was 1.6 – 2.3 higher than $R_{eco}$ estimated using EC-derived models. The daytime $R_{eco}$ model performs better the greater the influence of $R_{s\ Under-Tree}$

and the night-time $R_{eco}$ model and $R_s$ covaried more the higher the fraction of $R_{s\ Alley}$. We found values of $Q_{10} < 1$ in the vicinity of the olive tree and $R_{s\ Under-Tree}$ represented 39% of the $R_s$ of the olive grove. $CO_2$ pulses associated with precipitation events were detected, especially in the alley, during dry periods, and after extended periods without rain, but were not accurately detected by EC-derived models. We point out an interaction between several effects that vary in time and are different under the canopy than in the alleys that the accepted models to estimate $Q_{10}$ and $R_{eco}$ do not consider. These

results show a high spatial and temporal heterogeneity in soil respiration and the factors involved, which must be considered in future work in semi-arid agrosystems.

## 1 Introduction

Soil respiration ($R_s$) commonly refers to the natural release of $CO_2$ from the soil surface into the atmosphere and plays a key role in the carbon cycle. The global annual release of $CO_2$ through $R_s$ is ~ 95 PgC yr$^{-1}$ (Xu and Shang, 2016;

Zhao et al., 2017), which is approximately ten times higher than current emissions from fossil fuels (Friedlingstein et al., 2022). $R_s$ is the second largest carbon flux, accounting for 85%–90% of gross primary production (Hashimoto et al., 2015;





Jian et al., 2021). However, global $R_s$ is not constant but has been increasing by 0.04 PgC yr$^{-1}$ between 1960 and 2012 (Zhao et al., 2017). $R_s$ is influenced mainly by the carbon supply, temperature, and soil moisture (Hursh et al., 2017), and these parameters vary unevenly with global change. In fact, annual $R_s$ trends respond differently depending on latitude and biome,
increasing mainly in boreal zones and decreasing in tropical areas (Lei et al., 2021), whereas in semiarid regions such as the Mediterranean, no long-term trends are observed.

Mediterranean regions are at high risk of being impacted by climate change (Cramer et al., 2018). Although reduced rainfall in this region is expected to reduce Rs (Talmon et., 2011), the typical Mediterranean climate (irregular rainfall patterns, high evaporation rates, and water scarcity during summer months) can drive high temporal variation in $R_s$ because of its critical
sensitivity to soil moisture. Water controls the movement of soluble substrates when moisture is scarce, and of oxygen when it is abundant (Skopp et al., 1990). However, although $R_s$ has an important influence on the carbon cycle feedback in Mediterranean ecosystems, the understanding of $R_s$ in semiarid regions is still evolving (González-Ubierna and Lai, 2019) because the importance of water as a limiting factor on $R_s$ is more complex than previously thought (Leon et al., 2014).

Since heterotrophic $R_s$ is positively correlated with soil organic carbon content (Lei et al., 2021), which is generally low in
Mediterranean ecosystems (Munoz-Rojas et al., 2012), low values of $R_s$ are expected in such ecosystems. However, higher $R_s$ values are found in croplands in which carbon and water respond to management (Wollenberg et al., 2016). More productivity is expected in cropland because of the increase in nutrients provided by fertilizers, soil aeration, and irrigation. Therefore, the conversion to cropland of ecosystems typical of semi-arid areas can increase $R_s$ (Wang et al., 2023) compared with soils of natural ecosystems. However, the variety of agricultural systems in the Mediterranean is wide (Malek and
Verburg, 2017), which can translate into different responses of $R_s$ for crop types and management regimes.

One of the predominant tree crops in the Mediterranean basin is olives (*Olea europaea* L.). Their cultivation has significant economic, social, and environmental consequences for this region, which accounts for more than 90% of global production (FAOSTAT, 2022). Although allowing weed cover in alleys is widely accepted as sustainable crop management (Novara et al., 2021), weed growth is frequently controlled to avoid competition. A drawback of this practice is that the precipitation
regime promotes soil erosion in situations where the soil is bare (García-Ruiz et al., 2013), leading to a rise in soil $CO_2$ emissions. In addition, irrigation is a common practice in this crop during water scarcity periods when olive trees typically decrease their photosynthesis and, consequently, their yield during extended drought periods (Moriana et al., 2003). These different management options and inputs influence seasonal soil $CO_2$ emissions in Mediterranean olive agroecosystems (Montanaro et al., 2023) because they can affect the factors that control $R_s$.

Although soil temperature is the main driver of Mediterranean soil $CO_2$ emissions (González-Ubierna and Lai, 2019), water availability is a limiting factor. Therefore, the typical non-linear growth in $R_s$ as soil temperature increases is modulated by soil moisture in semiarid areas. Furthermore, the factor by which $R_s$ increases for every 10 °C rise in temperature, known as the apparent $Q_{10}$ (Davidson and Janssens, 2006) and frequently used to model $R_s$, is in turn influenced by different drivers in



semi-arid regions. However, the Birch effect (Birch, 1964) explains how carbon dioxide emissions change after the soil is
rewetted and has not been continuously explored in olive grove soils. Moreover, the expected alterations in precipitation
patterns may exert more substantial impacts on $R_s$ than projected temperature increases (Li et al., 2020). Therefore, water is a
critical determinant of $R_s$, and the techniques used to understand $R_s$ drivers in temperate climates (especially focused on
temperature) are not applicable in Mediterranean-type climates because they covary with soil moisture.

$R_s$ measurements are usually made at specific times, which makes it difficult to identify the drivers of $R_s$. In olive groves, $R_s$
has been studied using impedance measurements (Sierra et al., 2016), the respirometric method (Álvarez et al., 2007; Gómez
et al., 2009), gas chromatography (Marzaioli et al., 2010), process-based modeling (Nieto et al., 2013), or manual chamber
systems (Testi et al., 2008; Almagro et al., 2009; Bertolla et al., 2014; Turrini et al., 2017; Chamizo et al., 2017; Taguas et
al., 2021; Panettieri et al., 2022; Montanaro et al., 2023). The chamber system is widely used; however, in most cases,
measurements are performed on favorable weather days during (weekly or monthly) manual diurnal campaigns. Non-
continuous measurements have limitations regarding statistical replication, temporal dependency, annual budgets, and the
related level of uncertainty (Vargas and Le, 2023). It is necessary to generate precise long-term predictions of soil respiration
($R_s$) under varying environmental circumstances to enhance our understanding of its impact on $R_{eco}$ (Sánchez-Cañete et al.,
2017). Continuous measurements provide information at all temporal scales and can reveal phenomena that occur at times
when sampling is not usually performed, such as at night or during rain events, and can be key to understanding the
multitude of processes influencing $R_s$. In this sense, the eddy covariance (EC) technique has emerged as a significant tool,
enabling the assessment of ecosystem $CO_2$ vertical fluxes over extensive spatial and temporal scales, while preserving the
integrity of the studied ecosystem (Reichstein et al., 2005; Baldocchi et al., 2020). Because most ecosystem respiration ($R_{eco}$)
is due to $R_s$, it is common to use the Net Ecosystem Exchange (NEE) values measured with the eddy covariance technique to
model $R_{eco}$ as a proxy of $R_s$. However, the models used are limited because they include aboveground respiration and do not
consider the spatial heterogeneity of $R_s$ as well as the multitude of determinants involved in $R_s$ processes.

Variability in $R_s$ is not only temporal but also spatial (Stoyan et al., 2000), even for a "homogeneous" landscape system such
as a bare-soil olive grove. Although the spatial variability of $R_s$ in olive groves has been somewhat studied (Bertolla et al.,
2014; Montanaro et al., 2023), the difference between $R_s$ under trees versus alleys has not been studied before. In the vicinity
of the olive tree, we expected to find higher $R_s$ values because of autotrophic respiration of the roots and an increase in
heterotrophic respiration due to the contribution of photo substrates (Högberg et al, 2001). On the other hand, in the alley, we
expect to find lower $R_s$ because there is negligible autotrophic respiration or photo substrates. Therefore, the main objectives
of this study were to i) determine the temporal variability of $R_s$ in an olive grove; ii) study the differences between $R_s$ under
trees and in alleys over linear and angular transects; iii) analyze the main environmental drivers in $R_s$ and its temporal and
spatial dependence, including rain pulse events; and iv) assess modeled ecosystem respiration ($R_{eco}$) using data from an eddy
covariance tower and compare it with upscaled ecosystem $R_s$ of the olive grove obtained using an automatic multi-chamber



system. To address these objectives, we analyzed a full year's worth of soil and ecosystem respiration in an olive grove in southern Spain.

## 2 Material and Methods

### 2.1 Site description

This study was conducted in an irrigated olive grove (*Olea europaea* L." Arbequina") from "Cortijo Guadiana" (37°54´45´´N; 3°13´40´´W; 370 m.a.s.l.), in Úbeda (Jaén, Spain). Castillo de Canena, SL owns this traditional olive grove. The region experiences a Mediterranean climate (Csa; Köppen classification) with dry and warm summers, a mean annual temperature of 16 ºC, annual precipitation of 470 ± 160 mm, and potential evapotranspiration of 1205 ± 95 mm (n = 18; IFAPA, 2022). Between March and November, the olive trees received nocturnal drip irrigation three times per week (32 L
$h^{-1}$ for 8 hours). These trees were situated in clay loam soil and were subjected to fertigation, in which each tree received an additional 25–40 g of NPK fertilizer every night. The trees have an approximate height of 4 m, an age of ~85 years, a leaf area index of 1.89 ± 0.17 $m^2$ $m^{-2}$ and an estimated canopy radius of 2.8 ± 0.3 m. The plantation layout follows a 12x12 m frame, resulting in a tree distribution of approximately 69-70 trees per hectare, with 27% canopy cover (Data obtained with Google Earth and ImageJ software). In 2014, a homogeneous and flat parcel of the olive grove was selected for the
application of glyphosate-based herbicide in fall and winter to prevent plant growth. Since then, the soil of the plot has remained bare most of the time and extra herbicide has been applied to prevent rebound of the herbaceous cover and keep maximum control over external conditions. For soil characterization, see Aranda-Barranco et al. (2023).

### 2.2 Soil Respiration

In June 2020, six automatic soil $CO_2$ flux chambers were installed in an olive grove parcel treated with herbicide
over PVC manual collars (20 cm internal diameter) inserted into the soil at similar depths one week before starting the measurement. For volume correction, the average height of each collar was measured (n = 4) at the beginning of the installation. The system was composed of one IRGA (LI-8100A, Li-Cor, Lincoln, NE, USA) coupled to a 16-port multiplexer system (LI-8150, Li-Cor, Lincoln, NE, USA) with 3 opaque chambers (8100-104) and 3 clear chambers (8100-104C), the latter on bare soil. The observation interval was 2 min, and the pre-purge and post-purge time lengths were 30 and
45 s, respectively, for a whole cycle every 20 min. The multi-chamber system was configured to measure every 30 min to temporally match the eddy covariance and meteorological data. The data were downloaded monthly and processed with SoilFlux 4.2.1 Software to obtain a complete year of continuous measurements (~18000 flux values per chamber) by applying the best fit to concentration changes with time (76% exponential fit and 24% linear fit). $CO_2$ fluxes with a coefficient of determination ($R^2$) lower than 0.995 were discarded. Likewise, although the vegetation on the collars was
periodically removed, some $CO_2$ fluxes were discarded because of plant regrowth in some collars. A forward–backward predictor based on autoregressive moving average modeling (ARMA) in the time domain was used to fill the existing and



generated gaps (15% of the total dataset) and thus enable annual integration. For the rest of the analysis, only direct measurements were used.

To study spatial variability on $R_s$, three of these long-term chambers were placed under three different trees ($R_{s \text{ - Under-Tree}}$) at 0.8 m from the center of the olive tree and out of the fertigation drippers, while the other three were placed outside the influence of the olive trees, in the center of the alley, ($R_{s \text{ - Alley}}$) at 5.6 m from the epicenter of each olive tree (Fig. 1). All chambers were installed south of the tree, and the spontaneous seedlings of herbaceous plants were manually removed during each visit to the experimental site.



**Figure 1: Location of the olive grove in Spain, soil chamber distribution, area of campaign measurements, and eddy covariance tower position of the experimental site. © Google Earth 2023**

To study the spatial $R_s$, specific campaigns were conducted in two different setups. 1) To study the linear $R_s$ gradient between the tree and alley chambers, 15 additional collars were installed between long-term chambers (5 collars for each tree-alley location) to accommodate manual measurement campaigns. A portable IRGA (Li-7810 attached to Smart Chamber, Li-Cor, Lincoln, NE, USA) was used to quantify $R_s$ manually through 8 campaigns between September and



December 2021. 2) To study the angular $R_s$ gradient, 48 collars were installed surrounding the 3 selected trees (16 collars for each tree) and 9 campaigns were conducted during 2022 to quantify variations in $R_s$ concerning the orientation. To project $R_s$ to the ecosystem scale ($R_{s,eco}$), we weighted the alley and the under-tree $R_s$, first as a function of the ground and canopy cover, second as the average value of the alley and the under-tree linear gradient, and third as a correction for measuring in the south facing direction (See Supplementary Material). Simultaneous measurements of Li 7810 and Li 8100 showed a slope of 0.73 with $R^2 = 0.95$.

**2.3 Ecosystem Respiration.**

Throughout the study duration, ecosystem respiration ($R_{eco}$) was estimated from Net Ecosystem Exchange (NEE) measurements made within the olive grove employing the Eddy Covariance (EC) technique. An EC tower was set up in the center of the agroecosystem, with instruments positioned at a height of 9.3 m (5.3 m above the canopy). These instruments were used to monitor $CO_2$ levels and wind speeds at 10 Hz. Gas densities were measured using an enclosed-path infrared gas analyzer (IRGA, Li-Cor 7200; Lincoln, NE, USA). Simultaneously, wind speeds in the three vector components were recorded using a sonic anemometer (CSAT-3, Campbell Scientific, Logan, UT, USA).

EddyPro software version 7.0.8 computed the half-hourly NEE. Anomalies such as spikes, trends, dropouts, and abrupt variations in the eddy covariance data were filtered using the methodology outlined by Vickers and Mahrt (1997). Time lags between gas concentrations and wind speeds were compensated using covariance maximization. Half-hourly values of means, variances, and covariances were computed using the Reynolds decomposition rules. Double rotation of coordinates and spectral corrections for high frequency (Fratini et al., 2012) and low frequency (Moncrieff et al., 2006) were applied. Finally, the resulting fluxes were filtered according to the quality control method proposed by Mauder et al. (2013), and additional filters were applied to the half-hourly fluxes using the methodology described by Chamizo et al. (2017).

Approximately 48% of the data gaps in the agroecosystem measurements were attributed to missing data in the eddy covariance system, primarily stemming from adverse meteorological conditions, night-time stability conditions, instrumentation malfunctions, or quality control filters. We employed empirical modeling to fill in the missing data. Within the continuous eddy covariance database, we used the marginal distribution sampling technique (Reichstein et al., 2005) to replace missing values. This method is based on the replacement of missing values using a time window of several adjacent days. After replacing missing data, we applied two semi-empirical models to partition NEE into two components: gross primary production ($GPP_{eco}$) and ecosystem respiration ($R_{eco}$). The Reichstein et al. (2005) model $R_{eco - NT}$ is night-time based and it extends an exponential function of daytime respiration based on night-time data (with the assumption that $GPP_{eco}$ is negligible during night-time periods) to estimate daytime periods. The Lasslop et al. (2010) model estimates respiration ($R_{eco - DT}$) from fitting the Light-Response Curve during the daytime. Missing data replacement and partitioning were performed using the REddyPro R Package (Wutzler et al., 2018).





## 2.4 Environmental Measurements

Soil temperature ($T_s$) and soil water content (SWC) were measured at a depth of 5 cm near each chamber using a thermistor (LI- 8150-203, Li-Cor, Lincoln, NE, USA) and ECH2O model EC-5 soil moisture probes (Decagon Devices, Inc.,
Pullman, WA, USA). In addition, complementary environmental measurements were performed at the experimental site. The air temperature and relative humidity were recorded using a thermohygrometer (HC2S3, Rotronic, AG, Bassersdorf, Switzerland) positioned at a height of 5 m. The vapor pressure deficit (VPD) was computed using the data provided by the thermohygrometer. Incoming and outgoing components of short-wave and long-wave radiation were monitored using a four-component radiometer (CNR-4, Kipp and Zonen, Delft, Netherlands) positioned at a height of 7 m and situated 2 m away
from the tower. This setup allowed the determination of the net radiation and albedo. Incident and reflected PAR were also measured at 7 m using photodiodes (quantum sensor; Li-190, Lincoln, NE, USA). These meteorological data were sampled at 30-s intervals, averaged over 30-min periods, and subsequently stored in a data logger (CR3000, CSI).

## 2.5 Rain Pulse Events

The days between precipitation (PPT) events were counted to identify rain pulses. Intervals between PPT events
(hereafter inter-event periods, IEPs) were counted in days from the last PPT event, with a magnitude higher than 0.4 mm. The daily timescale was selected to avoid confounding diurnal $R_s$ variability and to achieve robust analyses. Once the event is reached, if there is rain on the following day, the IEP is reset to 1. The $R_s$ one day before the PPT event was taken as a reference. The $R_s$ event-response effect ($\Delta R_s$) was measured as the difference between the mean daytime $R_s$ post-event and the mean daytime $R_s$ pre-event, which is described as follows:

$\Delta R_s = R_{s\ post-event} - R_{s\ pre-event}$ ,                     (1)

The increase in the soil water content ($\Delta$SWC) was calculated analogously as in Equation 1. A pulse of rain was considered when the value of the difference of $R_s$ concerning the previous day was > 2.5 medians of the entire $R_s$ time series and coincided with a precipitation event. A potential fit was performed with the data excluding those whose value of the residual exceeds 3 $\mu$mol $CO_2$ m$^{-2}$ s$^{-1}$.

## 2.6 $Q_{10}$ calculations

Weekly windows were used to calculate $Q_{10}$. For this, an exponential adjustment was conducted according to

$$R_s = a e^{bT_s} ,\qquad\qquad\qquad\qquad\qquad\qquad\qquad\qquad (2)$$

To then calculate $Q_{10}$ as

$$Q_{10} = e^{10b} ,\qquad\qquad\qquad\qquad\qquad\qquad\qquad\qquad (3)$$



Where '$T_s$' is soil temperature (°C), 'a' represents the $R_s$ intercept at a soil temperature of 0°, and 'b' serves as the temperature coefficient, indicating the temperature sensitivity of $R_s$ and playing a role in the calculation of $Q_{10}$. (Lloyd and Taylor, 1994).

**2.7 Statistical Analysis**

We used 30-min values to characterize the diurnal variations, $Q_{10}$, and spatial gradients of $R_s$. Data at daily scales were used for the rest of the analysis, including a description of rainfall pulses, seasonal variability, and establishment of significant differences between trees and alleys. Polynomial curve fitting was used to optimize the relationship between ΔSWC variations as a function of the independent variables of precipitation and the previous SWC. The Shapiro–Wilk test determined the non-normality of the variables. The probability distribution of the variables was evaluated using the kernel density. Box plots and nonparametric statistical tests of two independent samples (Mann-Whitney test) were performed on the principal subsets of soil respiration, soil temperature, soil water content, and $Q_{10}$ to identify significant differences in the averages (3 chambers) of these variables. The annual balances were calculated as the sum of the daily values and the error was twice the square root of the accumulated variance of the standard deviation of the data. The graphs and statistical analyses were performed using Matlab (version R2020a).





# 3 Results

## 3.1 Seasonal variability in $R_s$ and environmental conditions

Significant differences were found between soil respiration ($R_s$) under trees and in alleys. Throughout the measurement year, $R_{s-Under\,Tree}$ was $11.5 \pm 3.8$ µmol $CO_2$ m$^{-2}$ s$^{-1}$ while $R_{s-Alley}$ was $4.3 \pm 2.3$ µmol $CO_2$ m$^{-2}$ s$^{-1}$, which means

2.7 times (Mann Whitney test; $p < 0.001$; $n = 17500$) more $R_s$ under the tree (Fig. 2a) than in the alley. The kernel density shows that the highest frequency of $R_{s-Alley}$ was found around 1.5 µmol $CO_2$ m$^{-2}$ s$^{-1}$ (due to the low winter values), whereas for $R_{s-Under\,Tree}$, it was close to that of the median (11.0 µmol $CO_2$ m$^{-2}$ s$^{-1}$).

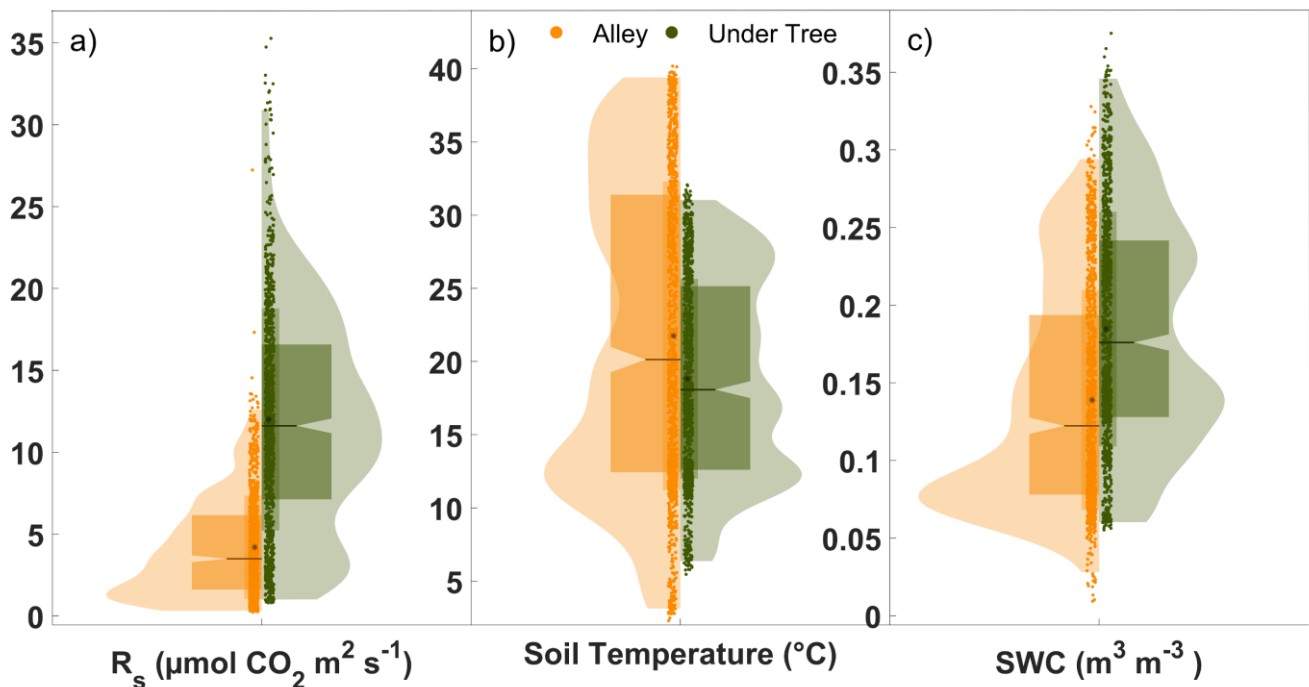

**Figure 2: Violin plots showing the daily averages of (a) soil respiration, (b) soil temperature, and (c) soil water content. Orange**
**and green refer to "alley" and "Under-Tree" measurements, respectively, each representing the average of three chambers. The**
**curve area is the kernel density, and the wide box represents the range q1–q3.**

Great seasonal variability in $R_s$ was observed. For both locations $R_s$ increased in the warmer months and decreased in the colder months (Fig. 3a), showing quite different values between months. Daily average minimums of $R_{s\,Alley} = 0.4$ µmol $CO_2$ m$^{-2}$ s$^{-1}$ and $R_{s\,Under\,Tree\,+} = 3.2$ µmol $CO_2$ m$^{-2}$ s$^{-1}$ were reached in January for both spatial locations. In contrast, the

maximum $R_{s-Alley}$ occurred in April (11.0 µmol $CO_2$ m$^{-2}$ s$^{-1}$) while the maximum $R_{s-UnderTree}$ occurred in May (23.9 µmol $CO_2$ m$^{-2}$ s$^{-1}$). The $R_s$ of both spatial locations were only similar in April, coinciding with herbicide application. In general, there was greater variability in the $R_s$ data under the tree ($\pm 3.8$ µmol $CO_2$ m$^{-2}$ s$^{-1}$; SD of data at 30 min) than in the alleys ($\pm 2.3$ µmol $CO_2$ m$^{-2}$ s$^{-1}$; SD of data at 30 min), which is visible in the entire daily time series (Fig. 3c).





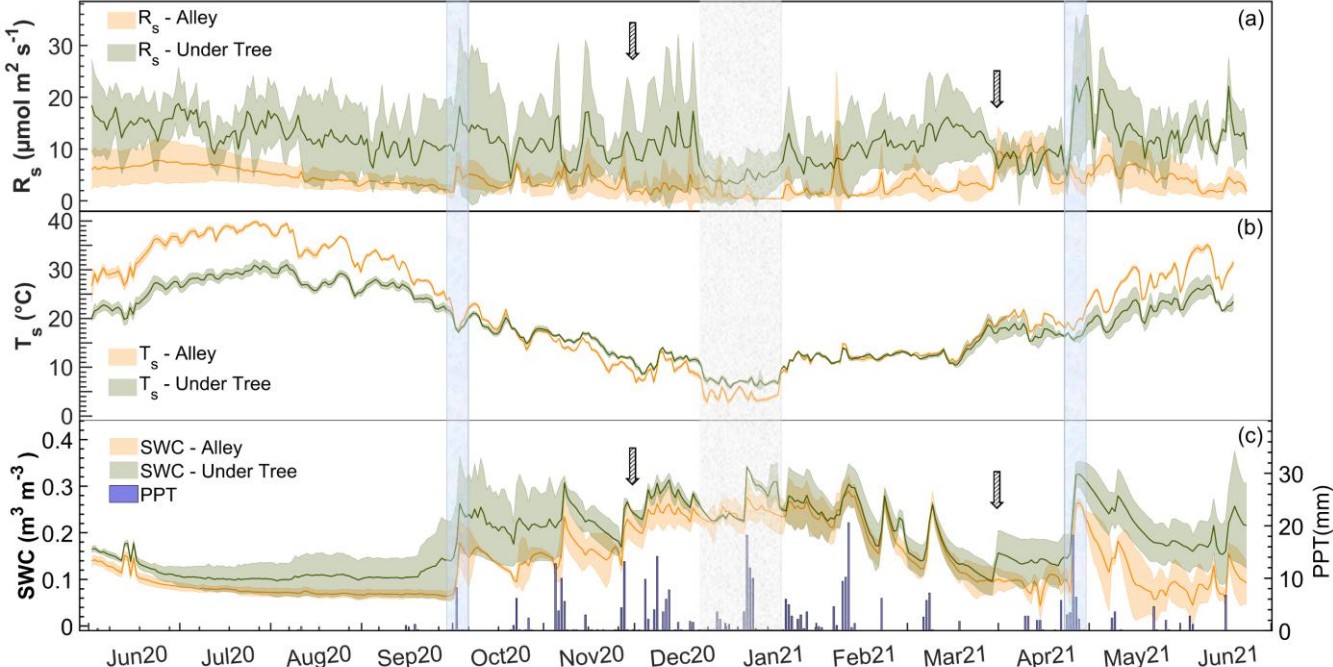

**Figure 3: Seasonal variation in daily averages under the tree and in the alley of a) soil respiration ($R_s$), b) soil temperature ($T_s$), and c) soil water content (SWC) and cumulative daily precipitation (PPT). Solid lines represent the mean of the three chambers, and the shaded area is the standard deviation. The first arrow indicates olive harvest, and the second arrow indicates herbicide application. The blue rectangles indicate two important rain pulses, whereas the gray rectangle indicates the period of the lowest values of $R_s$ and $T_s$.**

Differences were observed between soil temperature ($T_s$) and soil water content (SWC) under trees and in alleys (Fig. 2b,c). During the year of measurement, the average daily values of $T_{s – Under Tree} = 18.8 \pm 6.8$ °C; $T_{s - Alley} = 21.7 \pm 10.5$ °C, $SWC_{Under Tree} = 0.185 \pm 0.08$ m$^3$ m$^{-3}$ and $SWC_{Alley} = 0.139 \pm 0.07$ m$^3$ m$^{-3}$ so that $T_{s – Under Tree}$ was 13% lower (Mann Whitney test; $p < 0.001$; n = 17500) than $T_{s – Alley}$ and $SWC_{Under Tree}$ was 33% higher (Mann Whitney test; $p < 0.001$; n = 17500). However, $T_s$ y SWC showed large seasonal variation in the olive grove (Fig.3b and Fig.3c). In such a way, $T_{s – Under Tree}$ was higher than $T_{s – Alley}$ in the coldest months, showing a buffer effect of the trees on $T_s$. In general, $T_s$ was more variable in the alleys. More pronounced seasonal variability was observed in the alleys than under the tree, with $T_s$ maximum in July at 31.1 °C under the tree and 39.8 °C in the alleys, whereas minimum $T_s$ was 5.9°C under the tree and 2.8°C in the alley (January). From June to September, the high average daily $T_s$ coincided with the absence of precipitation. The end of the summer (September) presented the lowest daily SWC values of $SWC_{Under Tree} = 0.10$ m$^3$ m$^{-3}$ and $SWC_{Alley} = 0.04$ m$^3$ m$^{-3}$. (Fig. 3b). The first rainfall raised the SWC until it reached a maximum of $SWC_{Under Tree} = 0.34$ m$^3$ m$^{-3}$ and $SWC_{Alley} = 0.29$ m$^3$ m$^{-3}$





so that they became equal later (December). The return of irrigation in March again causes a difference in the SWC, which is maintained for the rest of the time series.

The annual precipitation was 322 mm falling mostly in autumn, winter, and spring (Fig. 3b), when 71 rain episodes were quantified with up to 27% of events with more than 4 mm day$^{-1}$, and a maximum of 21 mm day$^{-1}$. 48% of the events occurred on successive days (IEP = 1) and accounted for 53% of the accumulated PPT (Fig. 4a), while events with an interval without rain between 1 and 4 days and 5-9 accumulated the same precipitation (~ 50 mm) even though the former occurred more frequently. Precipitation led to a higher increase in SWC in the alleys than under the tree (Fig. 4b). However, these differences in ΔSWC were higher in small PPT events (< 4 mm), assuming an average ΔSWC of 0.003 under the tree and 0.007 in the alleys (Fig. 4b). Furthermore, this increase depended to a higher extent on the previous SWC and the amount of precipitation under the tree (R$^2$ = 0.6; Fig. 4c), whereas there was a lower relationship between the amount of precipitation and the previous SWC in the alley. (R$^2$ = 0.4; Fig. 4d).

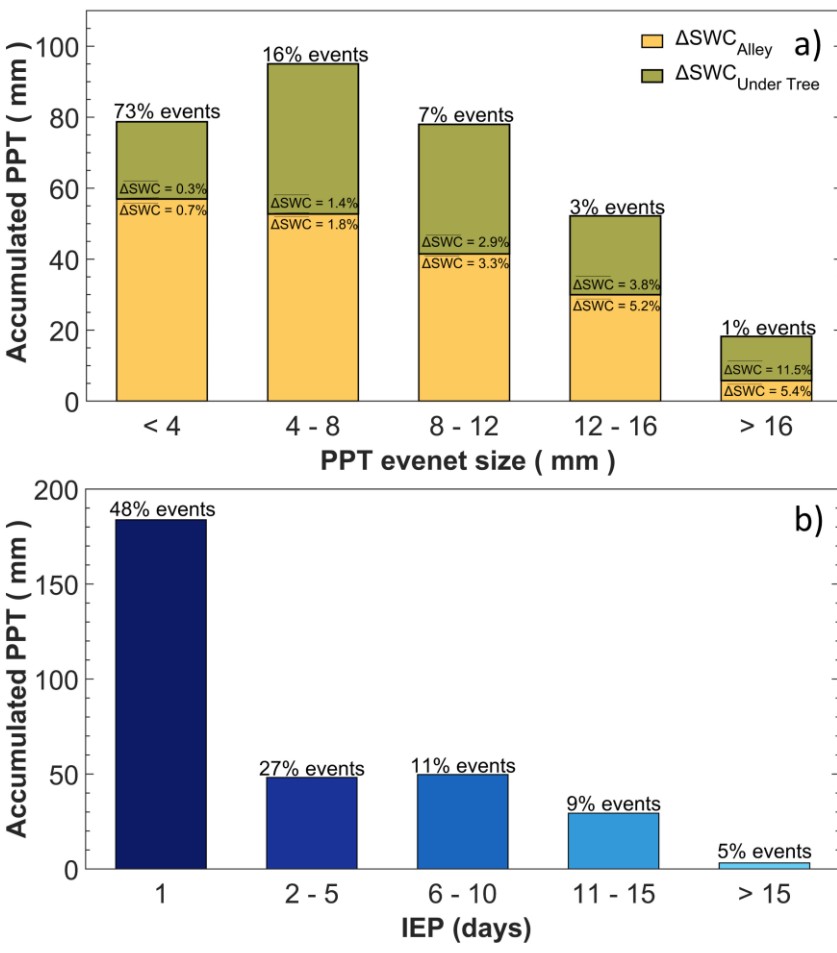





**Figure 4: Description of rainfall distribution. (a) Accumulated precipitation by inter-event period classes (IEP, days). (b) Accumulated precipitation by precipitation event size, where orange and green areas represent the proportion of increasing SWC**

**between alleys and under-tree. (c) Accumulated precipitation by the inter-event period of rain (IEP).**

### 3.2 Diurnal and spatial variability in $R_s$ chambers

In the alleys, we found daily variability with maximum $R_s$ at midday, coinciding with maximum temperatures (Fig. 5). However, the typical daily $R_s/T_s$ bell pattern was not always detected in the three chambers. During the weeks of July, one chamber showed no diurnal variability (Fig. 5b), not being an isolated case. In general, in winter weeks, diurnal variability

was detected with up to 3 µmol $CO_2$ m$^{-2}$ s$^{-1}$ more at midday than at night (data not shown). In spring months, $R_s$ was between 5 and 9 µmol $CO_2$ m$^{-2}$ s$^{-1}$ higher at midday versus night (Fig 5. d-f). On the other hand, the high variability of the fluxes under the trees caused the trend in $R_s$ to be statistically insignificant ($p_{value} = 0.57$; n = 240). However, we find an exception in the hottest summer months when soil respiration decreases in the afternoon while VPD increases. ($p_{value} < 0.05$; n = 240; Fig 5, a-c).

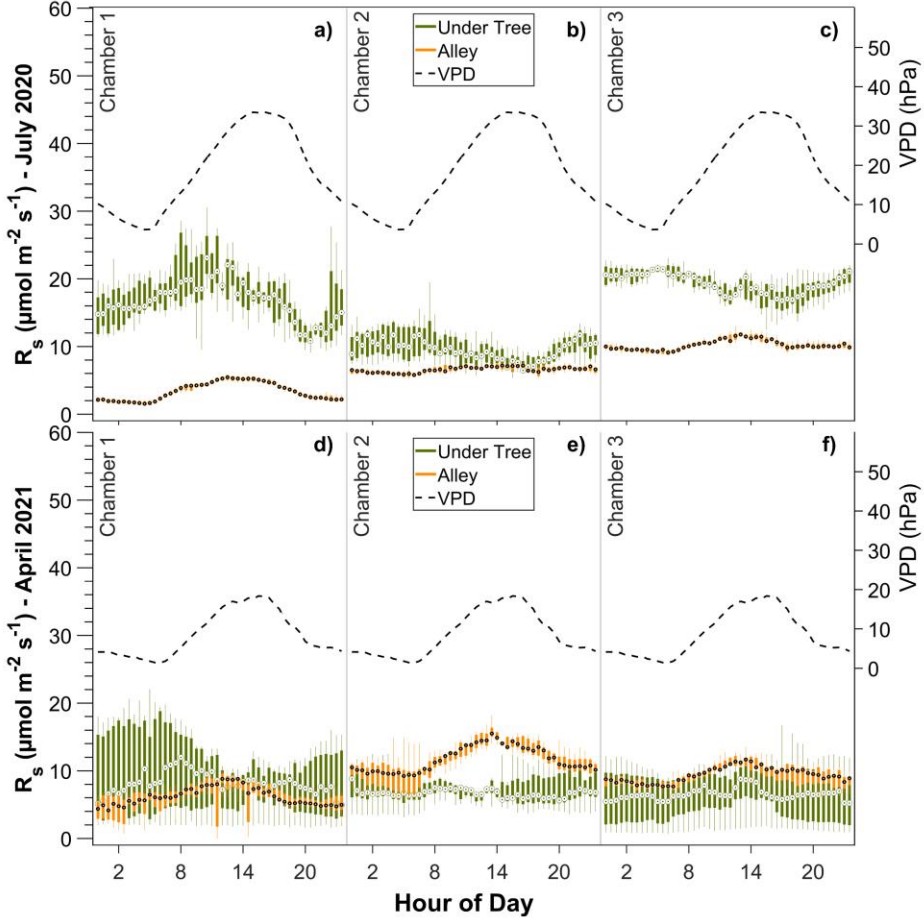






**Figure 5: Daily variability of $R_s$ in different periods. Each box plot represents 30-min data during a week. Inter-chamber variability is reflected in each compartment of each axis. Above: One week of data in July. Below: One week of data in April.**

Moreover, we found marked spatial variability in $R_s$ both under the canopy and in the alley. In the alleys, sometimes up to 3 times more respiration is observed in one chamber compared to another (Fig. 5; compare panels a and c), but these

differences between chambers, in turn, vary over time in such a way that a given chamber can sometimes measure the greatest and sometimes the least $CO_2$ emissions (switch in magnitude of chambers 2 and 3 between July and April; Fig 5; compare panels b/c and e/f). Finally, the ratio of $R_{s\text{-Under Tree}}$ and $R_{s\text{-Alley}}$ varied throughout the year. In the coldest months, although respiration decreased in both compartments, it was more noticeable in the alley so that magnitudes were reached under the tree of up to 7 times more than in the alleys (moving averages, Fig. 6a) when in addition the values of SWC in

both compartments were equal (Fig. 6c). Similarly, temperature variability on the ground was damped in the proximity of the olive tree so that the temperature was higher than that of the alley in the cold months and vice versa (Fig. 6b).

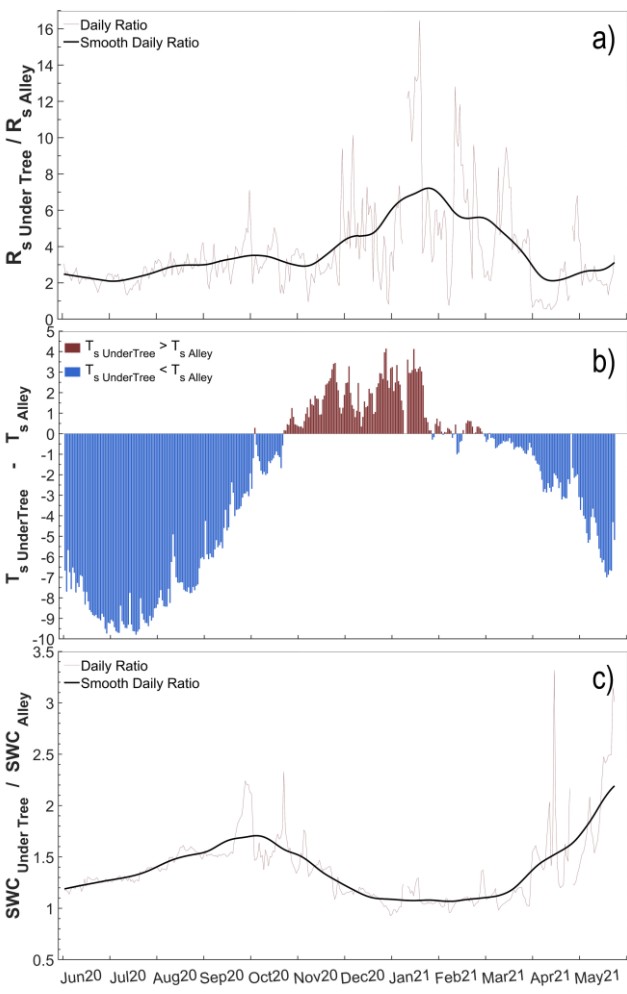



**Figure 6: Time series of a) $R_s$ ratios between chambers under trees and chambers in alleys. The gray dashed line refers to daily values and the black line refers to moving average daily values (± 30 adjacent days window); b) $T_s$ differences between chambers under the trees and chambers in alleys; and c) SWC ratios between chambers under the trees and chambers in alleys. The gray dashed line refers to the daily values, and the black line refers to the moving average daily values (± 30 adjacent day window).**

### 3.3 $Q_{10}$ variability

Seasonal variability was found in weekly $Q_{10}$ values, especially under the tree (Fig 7). In the alley, $Q_{10}$ ranged between 1.2 (warm months) and 2.0 (cold months), whereas under the tree, $Q_{10}$ ranged between 0.6 (warm months) and 1.8 (cold months). In general, the $Q_{10\ Under\ Tree}$ was higher and more variable than in the alley. Values for the entire study period were $Q_{10\ Alley} = 1.69 \pm 0.40$ and $Q_{10\ Under\ Tree} = 1.10 \pm 0.66$ (Mann Whitney test; $p < 0.01$). Hysteresis behavior was identified for $R_s$ data when plotted with both $T_s$ and SWC.

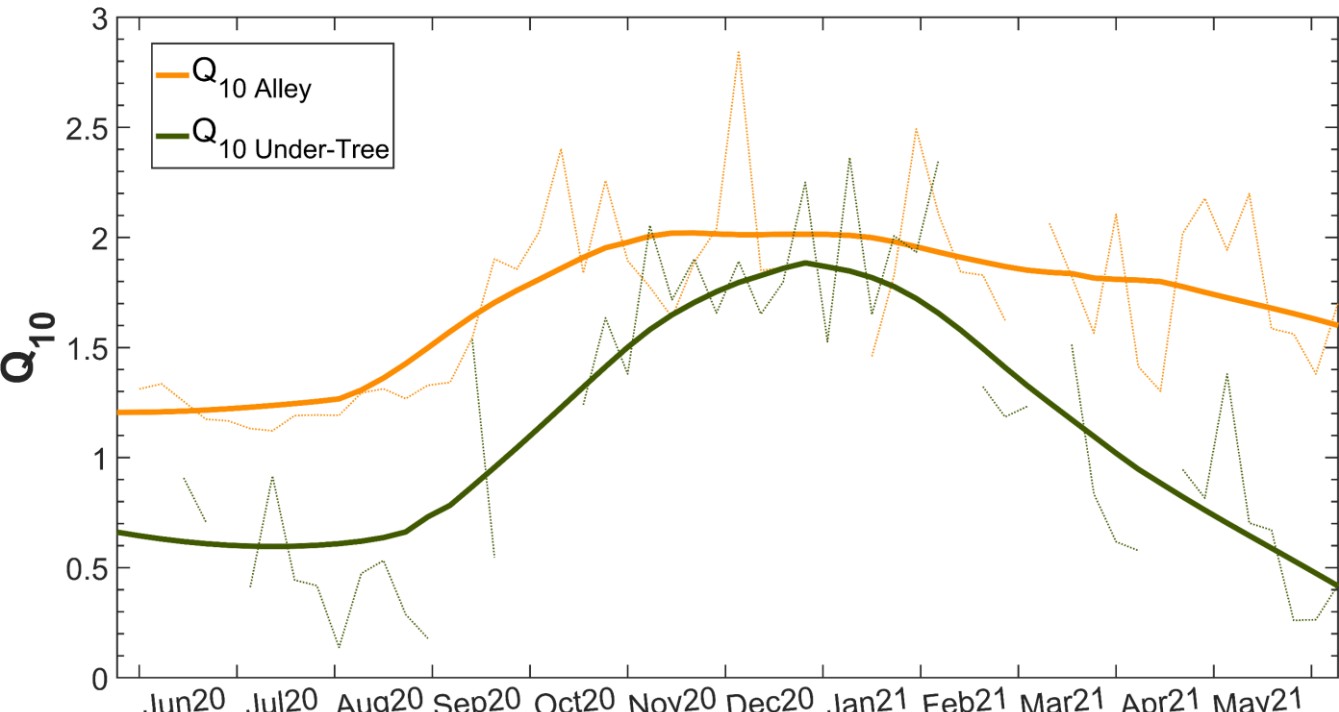

**Figure 7: Seasonal variation in $Q_{10}$ parameter in the alley and under-tree chambers. Dashed lines are the weekly $Q_{10}$ values, and solid lines refer to the moving average daily values (± 14 adjacent day window).**





### 3.4 Rain Pulses events: $R_s$ and $R_{eco}$

Of the seventy-five precipitation events, forty-one were accompanied by enhanced $R_s$. This was especially frequent when rain fell on dry soil, and the greatest $R_s$ rates followed the longest inter-event periods (IEPs), as shown in Figure 8a. In this way, notable rain pulses were detected when the IEP was large and SWC < 15% (Fig. 9a), whereas rainfall pulses were

scarce in months when the soil contained a moderate amount of water (> 15%). The relationship between $R_s$ and PPT was lost with SWC values higher than 15% in alleys and 20% under trees (Fig. 9b and Fig. 9c) both in the alleys and under the tree. The increase in $R_s$ with the appearance of a PPT event followed a nearly linear relationship in the alleys when SWC < 15%, whereas, under the tree, the increase was when SWC < 20%. The $R_{eco}$ (ecosystem respiration) values obtained via modeling appear not to respond to rainfall pulses, whether the soil was previously dry or not (Fig. 9d).

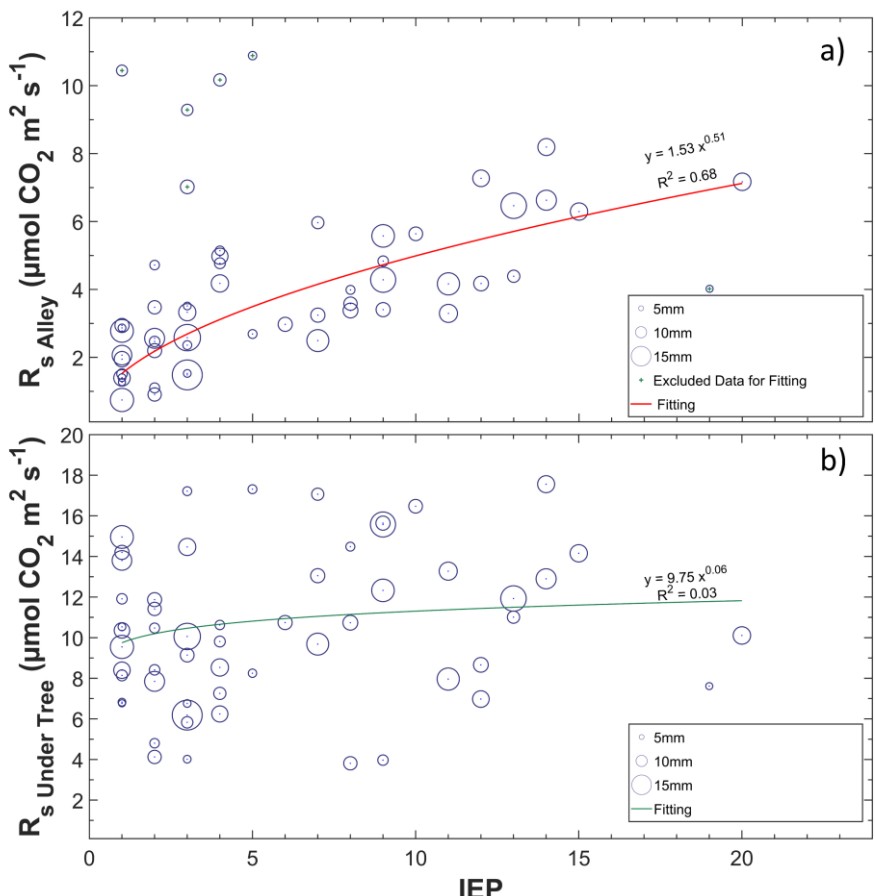


**Figure 8: Relationship between inter-event period (IEP) and daily soil respiration ($R_s$) in alleys (a) and Under-tree (b). The different sizes refer to the relative magnitude of the precipitation event, with the minimum being 0.4 mm and the maximum being 21 mm (daily values).**





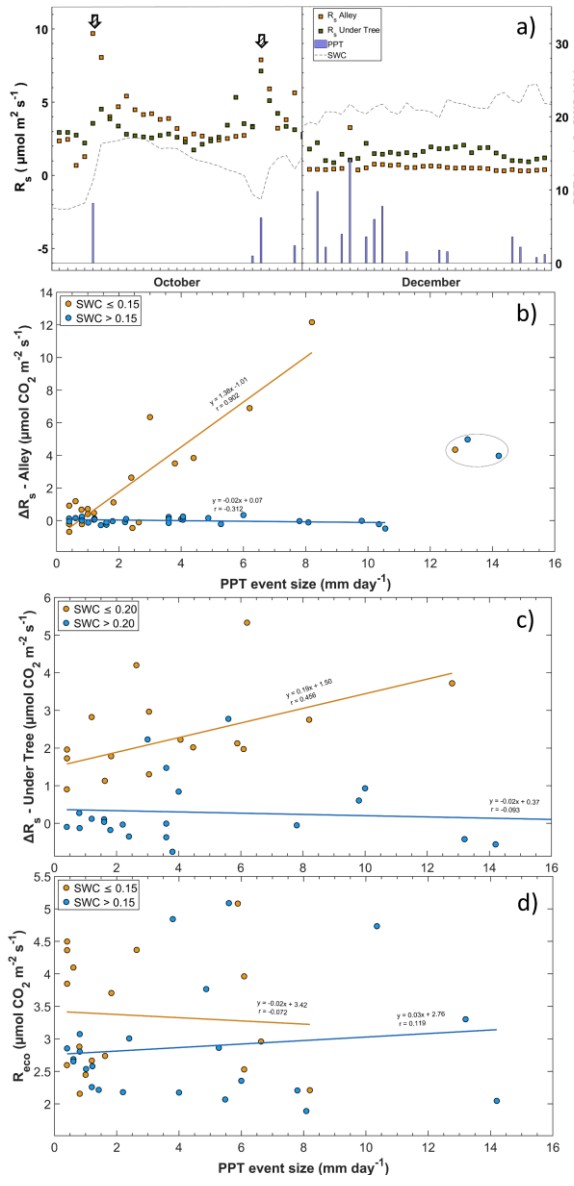

**Figure 9: Relationship between rainfall and soil respiration ($R_s$). a)** Rain pulses in a period with low soil water content (SWC) at the onset of the rain event (a; left) and a period with moderate SWC (a; right). Arrows indicate the moment of the pulse **b)** the relationship between the size of the PPT event (mm day$^{-1}$) and the variation in soil respiration rate ($\Delta R_s$; µmol $CO_2$ m$^{-2}$ s$^{-1}$) in the alley and **c)** under the tree. Measurements inside the circle are outside the fit. **d)** Relationship between the size of the PPT event (mm day$^{-1}$) and ecosystem respiration ($R_{eco}$; µmol $CO_2$ m$^{-2}$ s$^{-1}$). The lines represent linear regressions, and r is the correlation coefficient.



### 3.5 Spatial Gradients

Manual measurement campaigns revealed exponential transitions in both $R_s$ and $T_s$ in linear gradients from the tree to the alley (Fig. 10a and Fig. 10b). From 3.2 m away (4th collar from the tree), significant differences in $R_s$ and $T_s$ (Mann-Whitney test; $p < 0.05$) began to be found concerning the chamber closest to the tree trunk. However, from 3.2 m onwards, $T_s$ stabilized, while $R_s$ continued to decrease slightly. Because the average distance under the canopy from the epicenter of the trees was $2.8 \pm 0.3$ m, we can differentiate these two independent areas in terms of different $R_s$ behaviors. Thus, to project the value of $R_s$ to the ecosystem scale, two areas were considered where the average value used to weigh was the midpoint of the interpolation between sampled points (arrows in Fig. 10a). Regarding angular gradients, $R_s$ was higher on the south side than on the north side ($n = 27$; $p < 0.05$) during the sampling campaigns (Fig. 10c), and the temperature was higher on the east side (Fig. 10d) than on the north side ($n = 27$; $p < 0.05$). The high variability in SWC was due to punctual irrigation. Because the chamber was installed in the south, it was weighted according to these differences to scale $R_s$ up (Supplementary material).

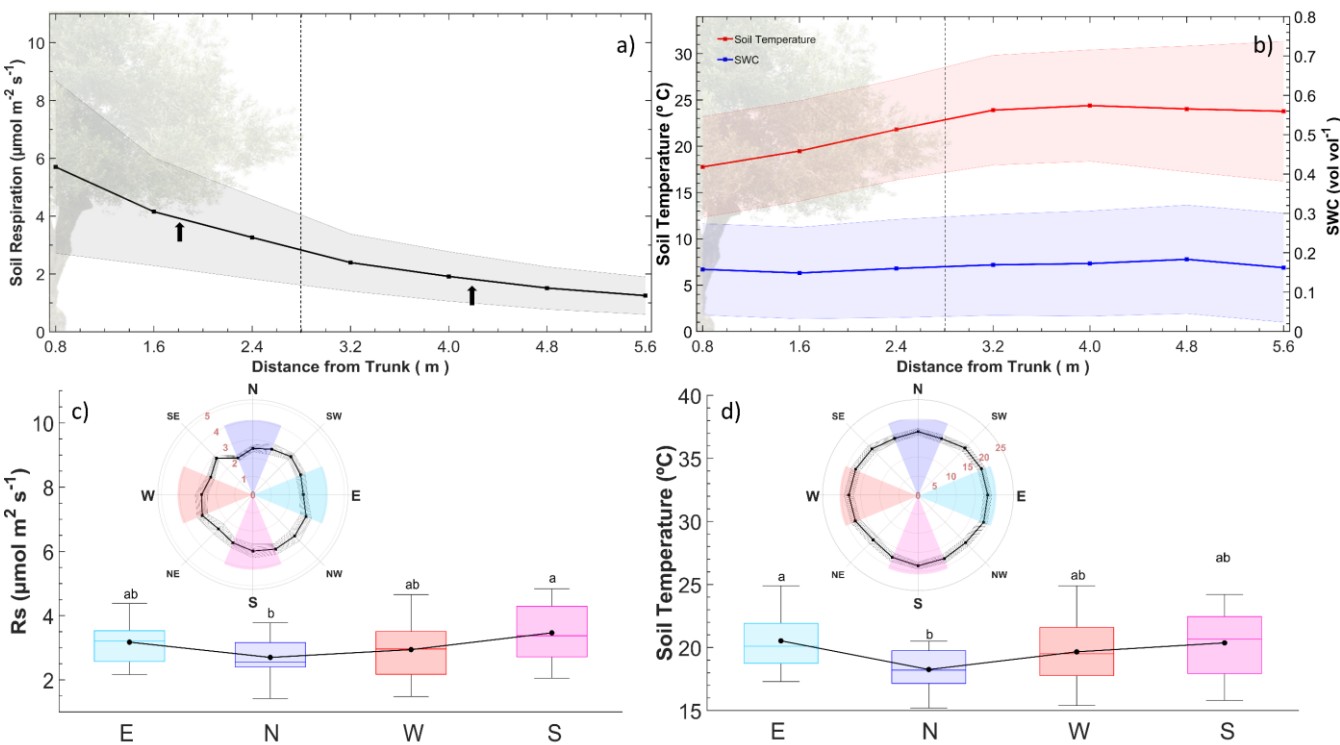

**Figure 10: Longitudinal and angular gradients of soil respiration ($R_s$), soil temperature ($T_s$), and soil water content (SWC). (a) $R_s$ measurements in a linear gradient from tree to alley collars. Each point represents the average (± standard deviation) of the manual measurement collar, and the line is a linear interpolation. The dashed vertical line refers to the separation between the**

**'Under-Tree' and 'Alley' regions of the transect (black arrows). The midpoint of the gradient of each region is considered as the weighting factor for $R_s$. ( b) $T_s$ and SWC variations in a linear gradient from the tree to alley collars ($n_{collar}$ = 8); (c– d) Angular gradient of $R_s$ and $T_s$ and differences between orientations. Each point represents a manual measurement collar, and the line**
**represents a linear interpolation. Areas in the angular graphic represent the cardinal grouping of the three measurements, and the box plot refers to the 9 campaigns (n = 27).**

### 3.6. Upscaled Rs,eco vs. modeled Reco.

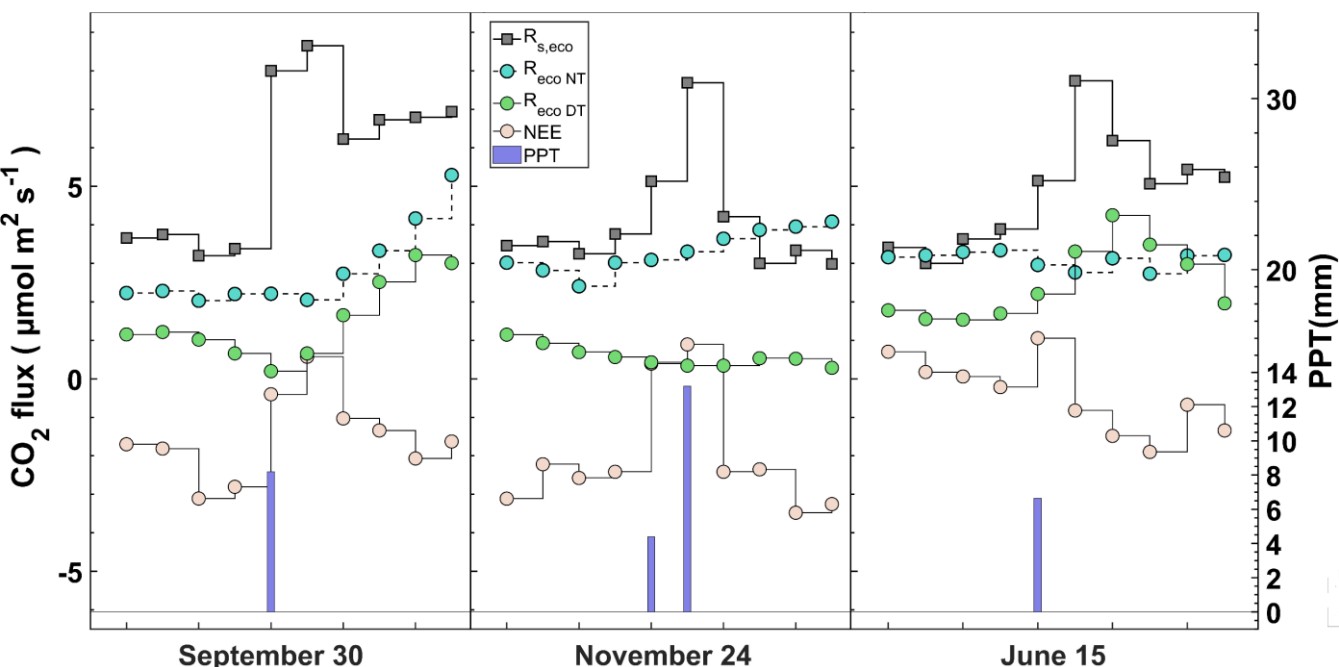

**Figure 11: Response of $CO_2$ fluxes (daily average) to a precipitation event at three different times in the time series. Net ecosystem**
**exchange (NEE), soil respiration upscaled to the ecosystem level ($R_{s, eco}$), night-time modeled ecosystem respiration ($R_{eco - NT}$) and day-time modeled ecosystem respiration ($R_{eco - DT}$).**

We show two approaches with the annually integrated $R_{eco}$ as the combination of measurement and empirical modeling based on EC data ($R_{eco-NT}$ =1310 ± 160 g C m$^{-2}$; $R_{eco-DT}$ = 850 ± 140 g C m$^{-2}$ ) and upscaling through chamber data ($R_{s,eco-}$ = 2100 ± 50 g C m$^{-2}$). During the warm months, the magnitudes of the chamber and night-time EC approaches are
quite different in magnitude, although they are consistent in the temporal variation. However, although the magnitudes are similar during the cold months, there is an inverse relationship between the two approaches. The daytime approach neither covaries nor has similar magnitudes with respiration data from chambers in hot months. On the contrary, the response of $R_{eco}$ $_{– DT}$ is similar to chambers the higher the influence of $R_{s-Under-Tree}$ on the ecosystem (Fig. 12a and Fig. 12c).




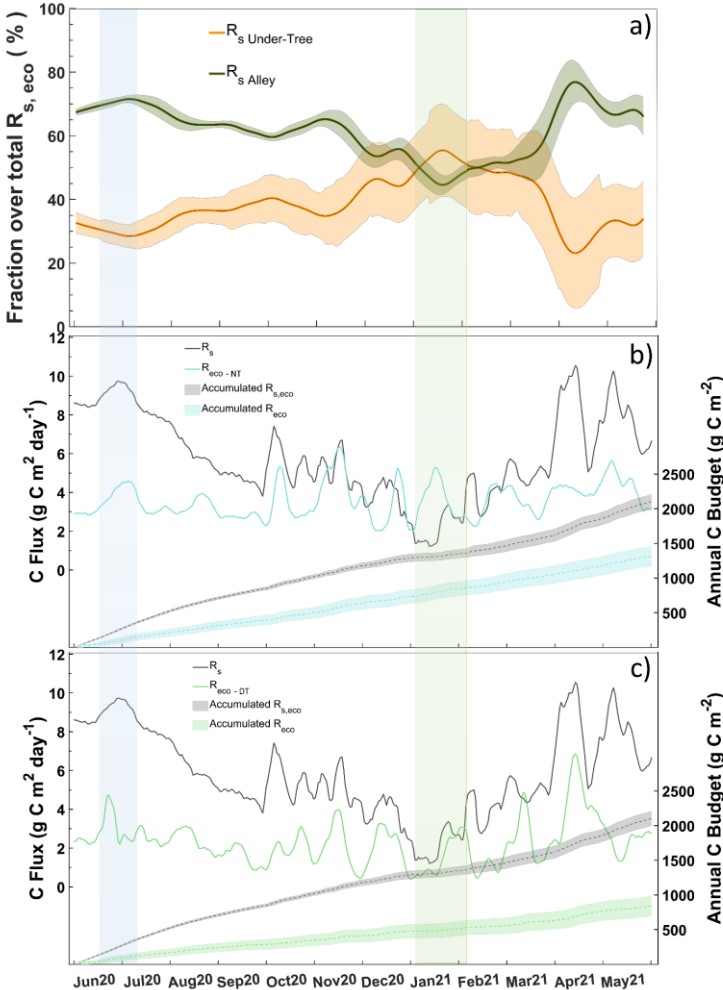

**Figure 12: a) Fraction of daily R$_{s\,Alley}$ and R$_{s\,Under-Tree}$ over estimated R$_{s,eco}$. b) Daily values of R$_{s,eco}$ and R$_{eco-NT}$ during the year (left) and the cumulative value of both (right). C) Daily values of R$_{s,eco}$ and R$_{eco-DT}$ during the year (left) and the cumulative value of both (right). The blue rectangle marks the warm period and high Rs alley influence. The green rectangle is the cold period and high R$_{s\,Under\,Tree}$ influence.**

## 4. Discussion

This study relies on a dataset spanning one continuous and complete year of respiration fluxes at soil and ecosystem scales and provides significant insights into the temporal and spatial variations of olive grove respiration as well as influencing factors. The abundance and continuity of half-hour measurements under trees and in alleys allow us to describe processes and trends that have not been described in olive groves by typical studies based on manual campaigns.



In this context, our findings affirmed a clear seasonal variability of $R_s$ and its main drivers (SWC, $T_s$), which is reflected in a
high range of values compared with other studies using chambers. Bertolla et al. (2014) and Testi et al. (2008) measured
daily values between 1.3 – 8.8 µmol m$^2$ s$^{-1}$ (n = 16) and 2.3 – 5.9 µmol m$^2$ s$^{-1}$ (n = 5; monthly) near the trunk, whereas our
study showed a wider range of 3.2 - 23.9 µmol m$^2$ s$^{-1}$ (n = 365; daily). Such differences could be partially explained by the
difference in the tree age (85 years for our individuals, versus primarily juvenile individuals between 2 and 7 years).
Juveniles will have less root development whose autotrophic and heterotrophic respiration is expected to be lower than that
of an adult individual. In our study, we can deduce a predominant influence of respiration associated with the roots on the
total soil activity, since $R_{s - UnderTree}$ exceeded on average three times that observed in the alleys on average. This excess
changed during the year, being between 5 and 15 times higher during the coldest months (Fig. 6a). Since the soil water
content was similar in this period (Fig. 6c), these differences could be due to i) heterotrophic respiration decreases in the
alleys due to the additive effect of a higher $Q_{10}$ (Fig. 7) and a higher decrease in the temperature versus under the canopy
(Fig. 6b); and ii) higher heterotrophic $R_{s - Under-Tree}$ due to a higher temperature compared to the alleys and differences in the
substrate due to the addition of root exudates and superficial leaf litter (Davidson and Janssens, 2006). The contribution of
heterotrophic respiration to total respiration is complex to estimate (Comeau et al., 2018). Therefore, considering that the
magnitude of $R_{s - Alley}$ during the winter was very small, we assumed that the contribution of heterotrophic respiration under
the tree is also small; therefore, the $R_{s Under-Tree}$ was largely controlled by rhizosphere respiration ($R_z$), which is the sum of
heterotrophic respiration linked to the root system and autotrophic respiration for maintenance and growth of the roots.

Despite the total canopy fraction "Under Tree" being only 27% in our agroecosystem, there are moments where the
proportion of $R_{s Under-Tree}$ contributes more than 50% to the $R_{s,eco}$ (Fig. 12a). Therefore, it cannot be assumed that the $R_{s Alleys}$
are representative of olive groves and most likely of mosaic tree crops such as savannas or 'dehesas'. In fact, in other
systems with an open area/canopy dichotomy (Tang et al., 2005), $R_{s Under-Tree}$ was an order of magnitude greater than $R_{s - Alley}$;
however, because the spatial gradients of $R_s$ were not quantified, the estimate at the ecosystem scale is unknown. In this
study, we quantified the gradient between the measurements taken under the tree and in the alleys and found an exponential
decrease as we moved away from the trunk. Therefore, although the influence of the roots extends throughout the crop, the
under-tree/alley dichotomy becomes significant around a 3-m separation. Therefore, for this experimental site, the canopy
radius (2.8m on average) can be a good proxy for determining the significant separation between the rhizosphere and alley.
Other studies have established a random collar sampling map (for instance see Turrini et al., 2017) with different separation
distances around the olive tree, which makes it difficult to integrate the role of rhizosphere respiration if correction distance
factors are not applied (Supplementary material). Moreover, we found more respiration on the south side of the arboreal
individuals, where the temperature was also higher. Although the campaigns, in which the gradient data were taken, only
covered 4 of the 12 months of the year, and spatial differences may also vary over time, we have used the data obtained to
weigh and scale the values of $R_s$ at the ecosystem scale and thus estimate $R_{s,eco}$.





In the estimation of model $R_{eco}$ with data derived from EC, we obtain an annual balance of 850 gC m$^{-2}$ (Daytime approach) and 1300 gC m$^{-2}$ (Nightime approach), whereas if we project $R_s$ at the ecosystem scale ($R_{s,eco}$), we obtain 2100 g C m$^{-2}$ (Fig. 12 b). The values obtained here with chambers may be similar to those found in grassland meadows (1990 g C m$^{-2}$; Bahn et al., 2008) and higher than other estimations in olive groves (860 ± 150 g C m$^{-2}$) although they were monthly or bimonthly

campaigns (Jian et al., 2021). A priori, $R_{s,eco}$ should be less than $R_{eco}$ because $R_{s,eco}$ is a fraction of $R_{eco}$ ($R_{eco} = R_{s,eco} + R_{AboveGround}$). However, the accumulated values obtained from the chamber ($R_{s,eco}$) are higher those obtained from the EC ($R_{eco}$), especially in summer. These differences between approaches could be related to the temporal mismatch between the two approaches. The chamber method takes snapshots of $R_s$, whereas EC models use a seven-day sliding window for their calculations. Furthermore, errors may be made in the EC models, such as the underestimation of nocturnal fluxes due to low

turbulence or the erroneous assumption that nocturnal respiration can be completely extrapolated to daytime respiration. On the other hand, during the day, we observe inverse relationships of $R_{s - Under Tree}/T_s$ to those that models based on EC usually assume ($Q_{10}$ values < 1 in Fig7) and that could lead to a bad estimate of $R_{eco}$ by the daytime method. Moreover, the greater the role of above-ground tree respiration on the $R_{eco}$, the worse the relationship between the $R_{s,eco}$, and $R_{eco - NT}$ (Fig. 12 a and Fig. 12b), which indicates that the widely accepted partitioning model based on Reichstein et al. (2005) does not apply

correctly to this ecosystem. It is expected that Lasslop et al. (2010) have greater agreement when the influence of the $R_{s\ Under\ Tree}$ is greater (Fig12) since $R_{eco}$ calculates it from the photosynthetic organisms' activity. Therefore, when a multi-chamber system is not available, we recommend for this type of ecosystem the use of daytime models in cold seasons and the use of nighttime models in hot seasons when most of the contribution to $R_s$ comes from the heterotrophic respiration of the alleys since the organisms will be little active in the transport of gases.


$Q_{10}$ differs in the alley and tree-base in terms of magnitude and seasonal evolution (Fig. 7). In the alley $Q_{10}$ was always higher than one, but under the canopy, we found periods with $Q_{10}$ values close to 1. This means that the variation in $R_s$ during this period is decoupled from changes in soil temperature. In addition, $Q_{10}$ values of one during the summer were inversely coupled, indicating that respiration decreased as temperature increased. That is, the respiration of the rhizosphere of the tree canopy was associated with soil temperature, contrary to what is expected. Therefore, the traditional parameter

$Q_{10}$, determined through field measurements of $R_s$ and temperature (Davidson and Janssens, 2006), cannot be used to define the respiration of Mediterranean ecosystems because of their large spatial and temporal heterogeneity. However, the $Q_{10}$ values were calculated in 7-day windows but could vary at different time scales. Currently, variations in $Q_{10}$ are controlled by soil and vegetation factors and not only climate (Chen et al., 2020). Therefore, although the global value of $Q_{10}$ is

estimated to be 1.5 (Bond-Lamberty and Thomson, 2010), the variability of reported $Q_{10}$ varies enormously, reaching values higher than 200 in ecosystems with very low temperatures (Mikan et al., 2002). Conversely, $Q_{10}$ values less than 1 have been found in other regions with semi-arid climates that include a dry period such as continental monsoon (Han and Jin, 2018), suggesting that it is not exclusively found in olive groves, but may be common in water-limited ecosystems with dry periods.





In addition, the regressions used to obtain $Q_{10}$ are usually not good when the water content in the soil is low (Wang et al., 2014).

The areas with the greatest uncertainty in global $R_s$ prediction models are semi-arid regions (Warner et al., 2019), where water acts as a limiting factor and $R_s$ decreases even with increasing temperature (Zhao et al., 2017). For example, in Mediterranean mountain grasslands, temperature is no longer a good predictor (Bahn et al. 2008). In our study, we observed a coincidence in the alley of the reduction of $R_s$ with the prolonged decrease in soil moisture in the prolonged summer drought (July - September), indicating a connection between $R_s$ and humidity (Fig. 3). Thus, even when the temperature increases in July, $R_s$ in the two compartments appears to decrease. However, $R_{s-Under-Tree}$ also decreased with the advance of summer even though $SWC_{Under-Tree}$ remained relatively constant due to irrigation. Also $R_{s-Under-Tree}$ was practically constant throughout the day except in summer when there is a negative relationship with VPD. This decoupling can be explained by changes in the photosynthesis of olive trees due to differences in VPD. It is known that $CO_2$ assimilation decreases with high VPD values when olive trees close their stomata (Fernández and Moreno, 1999), affecting the NEE (Chamizo et al., 2017). Stomatal closure has an impact on the rhizosphere because it inhibits the transport of photosynthetic products or carbohydrates from photosynthesis (which in turn depends on ecophysiological and meteorological factors) by the phloem, decreasing root activity and its exudates, thus decreasing respiration of the rhizosphere ($R_z$). Therefore, $R_z$ may be dynamically linked to vegetative growth, climate, or competition, all linked to $CO_2$ assimilation capacity of olive trees (Aranda-Barranco et al., 2023). Tang et al. (2005) established that the translocation time of photosynthetic products from leaves to roots can be between 7 and 12 h. This agrees with the $CO_2$ flux data shown in our study for the summer months (Fig. 5 a,c,d), where the peak decrease in $R_{s-Under-Tree}$ is between 4 and 7 h after the peak of solar noon temperature. The lag for the isotopic signal of photosynthesis in trees to appear at $R_s$ is in the range of days in other ecosystems (Ekblad and Högberg, 2001), which seems to indicate that stomatal closure would have an immediate effect (hours) on the reduction in $CO_2$ transport to the rhizosphere, whereas it would have a later effect (days) on the reduction of root exudates. Nevertheless, the association between photosynthesis and $R_{s-Under-Tree}$ can be confused with the relationship between $T_s$ and $R_s$.

Therefore, the response of $R_s$ to temperature fluctuations is influenced not only by soil temperature, making it crucial to consider additional factors such as SWC, photosynthesis, or precipitation events when modeling $R_s$ in Mediterranean environments (González-Ubierna and Lai, 2019). In our study, we see that the variations of SWC are higher in the alleys (Fig. 4a); therefore, the variability of the relationship between $R_s$ and $T_s$ will be higher if they are conditioned by humidity. In addition, the drivers are interrelated as temperature-dependent responses that are further influenced by soil moisture and precipitation (Hursh et al., 2017). Although the main transport process is molecular diffusion, rainwater can also produce an



immediate release of $CO_2$ by displacing gas within the pores of the soil, which can be the most important driver in terms of the seasonal trend of soil $CO_2$ efflux in semiarid ecosystems (Leon et al., 2014). We can see that water is a limiting factor for $R_s$ with a large rainfall pulse after the summer drought (first rectangle on the left; Fig. 3); however, in periods with high SWC, this relationship is lost (Fig. 9). Other studies in olive grove alleys (Testi et al., 2008; Sierra et al., 2016; Chamizo et al., 2017) have shown $R_s$ values between 0.5-1.6 µmol m$^2$ s$^{-1}$ ($n_{collar}-$ ~ 10) based on field campaign measurements taken

outside of rainy days. In contrast, using automatic measurements, we found a higher variability with 0.4 - 11.3 µmol m$^2$ s$^{-1}$ values (n = 365), but with a median of 1.5 µmol m$^2$ s$^{-1}$ (Fig. 2a), within the range of the other studies. This reflects the fact that continuous measurements can detect $CO_2$ pulses that tend to fall outside the usual ranges. The $R_{eco}$ (Eddy covariance) values obtained via modeling appear not to respond to rainfall pulses, whether the soil was previously dry or not (Fig. 10d), and this becomes evident when we observe how only the chamber fluxes respond to a PPT event. In our study, pulses of rain

were detected on 11% of the days of the year, which implied that up to 18% of $CO_2$ emissions occurred on days with pulses of rain. However, the variability in pulse length was high, with pulses lasting between 1 h and up to a day. Given that the duration of the pulses is usually between 3 and 6 h at our site, we can estimate that the total ecosystem contribution of rain pulses is less than 5%, as is the case in semi-arid areas (Delgado-Balbuena et al., 2023). At the ecosystem scale, we observed slight pulse signals with a lag of several days; therefore, the EC technique may not be the most appropriate for the

characterization of this phenomenon "in real time".

The effect of rain pulse events on $R_s$ is spatially dependent in agrosystems with two vegetation levels. The $CO_2$ pulses were higher with greater time elapsed since the last rain episode, but this relationship was only described in the alleys (Fig. 8), whereas the $CO_2$ pulses under the tree were less noticeable (Fig. 9). This could be because i) the $CO_2$ pulses are higher in

drier soils (Morillas et al., 2017), so the pulses may be less noticeable in irrigated areas, ii) the rain pulses are inhibited by the tree canopy, implying rain interception (in fact, we can see lower ΔSWC in rainy episodes under-tree) and/or iii) the priming effect is less noticeable when observing the computation of $R_z$ (autotroph + heterotroph), iv) the carbon supply and the different soil characteristics lead to different $CO_2$ release responses (Barnard et al., 2020). Rainfall pulses are reduced when vegetation cover is present (Liang et al, 2023) because the intensity of the event on the ground decreases. In general,

this region shows a paradoxical increase in extreme precipitation events, even as the total annual amount decreases (Zittis et al., 2021). Therefore, this phenomenon of releasing $CO_2$ could gain importance in the future in Mediterranean ecosystems. The implications of different management regimes in Mediterranean agroecosystems could be crucial for climate change mitigation strategies as they could lead to $R_s$ reductions (Wollenberg et al., 2016; Montanaro et al., 2023) with the use of covers that reduce losses of $CO_2$ from precipitation events.




Because the temperature and SWC are relatively constant in the alleys, the inter-variability of Rs_alley could be due to changes in the communities and activities of the microbiota present, which depends on the microscale distribution of microaggregates, nutrient levels, pH, oxygen availability, or substrate availability, which are factors that regulate soil microbial communities and activities (Wilpiszeski et al., 2019; Hermans et al., 2020), in addition to temperature and

humidity. In addition, the heterogeneity detected under the tree may be conditioned by the unequal distribution of the canopy (high standard deviation in $T_s$), leaf debris, root system, and position of drippers, as well as the dissolution of $CO_2$ in soil water or root xylem water (Bloemen et al., 2013). Moreover, transformation into bicarbonate ions in these high-pH soils or the dissolution and precipitation processes of carbonate minerals (Angert et al., 2015) can cause temporary decoupling between soil–gas exchange fluxes and biological $R_s$ (Xu and Shang, 2016).

**5. Conclusions**

Continuous measurement with a multi-chamber system revealed a higher range of soil respiration values than those previously reported in olive groves. $R_{s - Under\ Tree}$ was on average 3 times higher than $R_{s - Alley}$, especially in the cold months when 50% of $R_s$ at the ecosystem level came from $R_{s - Under\ Tree}$, even though the canopy fraction represents only 27%. Therefore, it cannot be assumed that $R_{s – Alley}$ is representative of olive grove soil respiration. Also, consistent patterns

showing higher $R_s$ on the south side of the tree individuals and exponential decrease from the trees to the alley center allowed us to calculate the accumulated $R_s$ at the ecosystem level.

The annual accumulation was 2100 g C m$^{-2}$ and twice the ecosystem respiration ($R_{eco}$) obtained from the eddy covariance. The higher the role of tree respiration on the $R_s$ of the ecosystem, the worse the relationship between the $R_s$ behavior of the chambers and the modeled $R_{eco- NT}$, showing that temperature-based models are insufficient in olive groves in cold months.

Furthermore, inverse relationships between $R_s$ and temperature were found in summer ($Q_{10}$ less than 1), indicating that the variation of $R_s$ during this period is decoupled from changes in soil temperature and may be directly related to stomatal closure with high VPD and transport of newly produced photosynthates.

Finally, large pulses of $CO_2$ were observed when rain fell on dry soil and were higher with longer rain-free periods. The tree structure reduced the relationship and magnitude of the pulses with precipitation, thus reflecting interception. The pulses

were determined by the previous humidity conditions, and the detection of the pulses was lost when autotrophic and heterotrophic respiration were observed together. The continuity of the measurements allowed clear spatial differences to be established in the response of $R_s$ to changes in temperature, humidity, and rainfall pulses. All these findings show spatial and temporal variability in $R_s$ and its drivers that should be considered in future studies of soil $CO_2$ respiration in Mediterranean agrosystems.



## Declaration of Competing Interest

The authors declare that they have no known competing financial interests or personal relationships that could have appeared to influence the work reported in this paper.

## Acknowledgments

This work was supported by the Spanish Ministry of Science and Innovation through projects CGL2017- 83538-471 C3- 1-R (ELEMENTAL) including European Union ERDF funds [grant number PRE2018-085638], the PID2020-117825GB-C21 and PID2020-117825GB-C22 (INTEGRATYON3) and the projects ICAERSA (P18-RT-3629), OLEAGEIs (B-RNM-60-UGR20) and MORADO (C-EXP-366-UGR23) funded by the Andalusian regional government and the European Union including European Founds for Regional Development. A-B. S acknowledges support from the FPU grant by the Ministry of Universities of Spain. [REF:]. FPU19/01647. Thanks are given to the Group of Castillo de Canena for the use of their farm as an experimental site and their people for continuous cooperation and to Manuel Martos for collecting data in the manual campaigns with the portable chamber.

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
