# Peer review of "Spatial and temporal heterogeneity of soil respiration in a bare-soil Mediterranean olive grove"

_EGUsphere, 2024_

## Referee Comment (RC1)

**General comments**

This manuscript presents valuable measurements of soil respiration in a semi-arid agricultural setting. In general, the work highlights many of the challenges in measuring and interpreting semi-arid soil respiration. The authors demonstrate a strong grasp of the literature relevant to semi-arid olive plantations, and the potential competing factors influencing soil respiration. Unfortunately, the organization of the introduction and discussion does a disservice to the data presented. The topics introduced range widely, with the four stated objectives (lines 92-96) ranging widely from presenting specific data, to interpreting driving mechanisms, to comparing eddy covariance models. The manuscript feels caught between exploring soil mechanisms on one hand, and upscaling to the ecosystem level on the other. The comparison between chamber measurements and eddy covariance, while in principle a relevant topic, is underdeveloped, even to the point of distraction from the main contributions of the manuscript. Conversely, the interesting scientific contributions on the driving mechanisms of soil respiration are understated in the scientific arc of the manuscript. These should be further developed with analysis, potentially leading to much more satisfying conclusions on the role of heterotrophic vs rhizosphere respiration, and also the interplay between moisture and temperature. Additionally, I would be particularly interested in further discussion of mechanisms by which the tree itself drives factors that influence Rs. I suggest the acceptance of this manuscript with minor revisions to improve the scientific organization and to further the analysis and interpretation of potentially enlightening data.

**Specific comments**
- 19-20: The mention of EC model performance feels out of place in the abstract, especially since the daytime and nighttime models are not otherwise mentioned in the abstract. This is emblematic of the rest of the manuscript, where the EC contribution is potentially interesting, but not developed sufficiently to address scientific questions.
- 37-50: The discussion of mediterranean climate, climate change, and drivers of Rs is hard to digest all at once. I suggest introducing Rs drivers, the mediterranean climate, and potential climate change as separate topics.
- 64: The Birch effect is introduced in name, but not in terms of mechanisms relevant to the paragraph - further discussion of mechanisms, such as the Birch effect, would greatly increase the applicability of this research.
- 69 and elsewhere: the power of semi-continuous measurements vs measurement campaigns is apparently true, but not sufficiently developed here. The discussion of maximum variability in the discussion section is not necessarily convincing. It would be nice to see some analysis or at least discussion on how continuous measurements improved either the understanding of mechanisms or the upscaling efforts.
- 92-94: These aims feel both too wide and too overlapping. I think the entire manuscript could be strengthened significantly by focusing these aims, and adjusting the introduction and discussion accordingly.
- Rain pulse events: Section 2.5, 3.1, and 3.4. The role of rain-pulse events becomes one of the main themes of the paper, but it is relatively unmentioned in the introduction. The concept should be better introduced in the introduction, which would also strengthen the discussion and interpretation of mechanisms. Particularly, I would be interested to read

the author's discussion on the role of mechanical porespace displacement vs increased respiration. I also thought the discussion of types of rain events was excessive, e.g. Figure 4 and lines 248-257.

- Section 3.3: Q10 variability is discussed using only the Ts and SWC data from 5cm depth. Hysteresis is mentioned, but further discussion is warranted on the role of time lags for temperature and water to propagate through soil. This is relevant also for the discussion of rain pulses.
- Tree-alley dichotomy: The data very nicely examines the gradient of temperature, moisture, and Rs going between the tree and the alley. Especially given how nice the data appears, this seems like it would be a great opportunity to investigate the mechanistic role of the tree on Rs and its drivers. I was also wondering about why it was necessary to identify a tree-alley dichotomy, when it appears so nicely represented as a continuous transition.
- Figure 5 and Discussion 430-447: The discussion of VPD and plant-Rs relationships is interesting, but not necessarily convincingly supported by the data shown in figure 5. This would feel better supported if the role of root exudates were introduced as a potential mechanism in the introduction.
- Figure 6, and elsewhere: Why are Rs and SWC reported as ratios between the two locations, while temperature is the simple difference? This was also bothering me elsewhere in the manuscript, where ratios may not have been appropriate, as in comparing Rs spatial variability when all the emissions are quite low.
- 391-409: This section particularly feels underdeveloped. It is apparent that neither model is perfectly representing the ecosystem, but the mechanisms are unclear. I feel it is premature to suggest using the flawed models, perhaps it would be better to identify this as an area requiring further research on modeling mechanisms.
- 468-470: The discussion of why EC did not pick up the Birch effect is quite interesting, but it is difficult to think why EC would not be real-time. Perhaps this is also best left as an area for future research.

**Technical corrections**
Generally here are some paragraph-level organizational comments:
- Lines 20-21, those are two relatively big and unrelated findings that don't fit well together in one sentence.
- 31: Not clear what is set of carbon fluxes is being referenced with 'second largest'
- 239: 'y' should be translated to 'and'
- 267: Not clear what trend "the trend in Rs" is referring to
- 280 and elsewhere: "compartments" may not be the right word to refer to different chamber settings.
- 405: It is expected that lasslop et al. (2010) would have greater…
- 486: Rs_alley should be subscripted

---

## Community Comment (CC1)

This detailed assessment of the spatial and temporal variability in soil respiration in an irrigated olive orchard in southern Spain provides insights into the biotic and abiotic controls over this important component of the carbon cycle. Continuous fluxes were measured under trees and in the alleyways between trees for one year, and more detailed spatial measurements were taken on a campaign basis to evaluate the effects of distance from the tree trunks and directional position around the trees. The research is important because semi-arid and Mediterranean systems are under-represented in ecosystem research but are potentially more sensitive than other systems to changes in climate (warming, drying) and land management (such as irrigation, herbicide). Moreover, the analysis of spatial variations due to the spacing of the trees allowed for detailed assessment of the contribution of vegetation versus bare ground to the fluxes, and these were scaled up for comparison to ecosystem respiration measured at a nearby eddy covariance tower.

A substantial number of analyses were conducted with the large dataset and were presented clearly in the figures. The main findings of the paper were that the respiration fluxes from under the trees was about three times higher than from the alleyways, on average per m2, whereas the total contribution from under the trees to soil respiration was 39% scaled to the whole area. Unfortunately but not unsurprisingly, the scaled-up soil respiration was substantially higher than the ecosystem respiration measured from the flux tower. The temperature sensitivity estimated as Q10 on a weekly basis from the soil respiration fluxes was higher in the alleyways than under the trees during the hotter and drier summer period but not during the wetter cool season.

The paper could benefit from considering a few questions and suggestions. The organization of the main findings could be streamlined a bit and the authors should consider whether all the figures are really necessary to support the take-home messages. My main concern is that the role of soil water content (SWC) and its regulation of microbial carbon substrate availability could be investigated in a bit more detail.

1) It was not clear how the irrigations in summer affected SWC and Rs under the trees. Are the irrigations shown in Fig 3? If so they are hard to see. Consider changing the monthly indicator tics on the x-axis to point downwards instead of upwards.
2) I would expect that infrequent, large precipitation events would have a disproportionate influence over the Rs. Did you look at this? Can you do an analysis of delta Rs for the different PPT event sizes in the same way as in Fig 5a? Or plot delta-Rs versus delta-SWC by bin. You did some plots of delta-Rs versus event size in Figure 9 for a couple example weeks; why not for the full dataset? Maybe the effect of inter-event period is more important than the event size in regulating the delta-Rs across the full dataset, in which case it would be good to make that more clear.
3) I don't get much out of Fig 6, consider moving it to supplement unless it is critical to one of your main findings.
4) Why did you not consider the effect of SWC on temperature sensitivity? There is a large literature on this and it seems like a missed opportunity not to incorporate an alternative analysis that would allow it.
5) Related to the point above, the apparent Q10 values <1 are not biologically meaningful, so there must be an artefact. Why would Rs increase with decreasing temperature, only under the trees? Perhaps this is the result of the night-time irrigations stimulating Rs when the temperatures are lowest? Maybe the results would be different if you considered only the midday soil temperature and Rs, or filter the data for SWC such as the bins in Fig 9 (why were different SWC thresholds used for alleyways and under trees?).

6) Is there any data available on soil, root or microbial carbon stocks under the trees and in the alleyways? If so these could be used to improve the discussion of the biotic regulation of the fluxes via rhizosphere processes. I can understand if the authors prefer not to speculate too much, if insufficient data is available.

A secondary important consideration is that the Reco partitioning using the standard daytime and nighttime methods does not capture these important $CO_2$ pulse responses to precipitation events.

7) What percentage of annual Rs is released during those pulses for the alleyways and under trees, and scaled to the ecosystem? I believe this could be calculated from the accumulated delta-Rs. Is this approximately similar to the magnitude of difference between scaled-Rs and Reco?

8) I think it's good that you have called the eddy flux data "modelled" Reco but it still could be viewed as evidence against the whole eddy covariance method. In the abstract and the text it would be helpful to make it really clear that the partitioning method is the issue, not necessarily the data.

9) If there is time to do additional analyses consider using a neural network partitioning method that includes soil moisture (and perhaps VPD), not just temperature, in the eddy flux estimates of Reco, or perhaps using only actual nighttime quality-controlled data for both Rs and Reco for a more direct comparison. But that might be beyond the scope of the current work.

Specific comments:

Section 2.5 and throughout. Please clarify the language related to "rain pulse events" or "pulse events" because it is somewhat confusing. Instead consider calling them "$CO_2$ pulses" in response to "precipitation events".

---

## Author Comment (AC3)

[revised manuscript text omitted]

65  in semi-arid regions. Q$_{10}$ is rarely measured continuously and obtaining this parameter uninterruptedly could elucidate the existence and interdependence of more $R_s$ drivers. Additionally, rain-pulse events play a key role in semi-arid regions (e.g., Mediterranean) during dry seasons, and may alter the annual carbon balance. The Birch effect (Birch, 1964) describes how carbon dioxide emissions increase by a high rate of rapid mineralization after the soil is rewetted due to a rain pulse event. . This mechanism can contribute to 5% of total annual respiration in semi-arid regions ( Delgado-Balbuena et al., 2023), can

[revised manuscript text omitted]
 (Table 1). For estimation of canopy coverage, an average of 50 canopy areas were measured using Google Earth snapshot and imageJ Software. To know the part of the linear gradient that belongs to the tree and alley another 50 canopy lengths (each one as the average of four cardinal measures) were measured and averaged. The results shown in Fig 10 (arrows) were used to know the average value of the 'Under-Tree' and 'Alley' linear gradient. Simultaneous measurements of Li 7810 and Li 8100 showed a slope of 0.73 with $R^2 = 0.95$.

| Table 1 Correction and weighting coefficients for the homogeneous calculation of $R_{s,eco}$ (See Fig 8) | | | |
|---|---|---|---|
| | Weighting by Canopy/Alley Coverage (%) | linear distance correction factor | Orientation correction factor |
| Alley | 73 | 1.2 | NA |
| Under-Tree | 27 | 0.7 | 0.9 |

[revised manuscript text omitted]
 July, one alley chamber consistently showed no diurnal variability. In general, in winter, diurnal variability was detected with up to 3 $\mu mol\ CO_2\ m^{-2}\ s^{-1}$ more at midday than at night (data not shown). In spring, $R_s$ was between 5 and 9 $\mu mol\ CO_2\ m^{-2}\ s^{-1}$ higher at midday versus night (Fig 4. d-f). On the other hand, the high variability of the fluxes under the trees caused the daily trend in $R_s$ to be statistically insignificant ($p_{value} = 0.57$; n = 240). However, we find an exception in the hottest summer months when soil respiration decreases in the afternoon while VPD increases. ($p_{value} < 0.05$; n = 240; Fig 5, a-c).

[Figure]

**Figure 4: Daily variability of $R_s$ in different periods. Each box plot represents 30-min data during a week. Inter-chamber variability is reflected in each compartment of each axis. Above: One week of data in July. Below: One week of data in April.**

Moreover, we found marked spatial variability in $R_s$ both under the canopy and in the alley. In the alleys, sometimes up to 3 times more respiration is observed in one chamber compared to another (Fig. 4; compare panels a and c), but these differences between chambers, in turn, vary over time in such a way that a given chamber can sometimes measure the greatest and sometimes the least $CO_2$ emissions (switch in magnitude of chambers 2 and 3 between July and April; Fig 4; compare panels b/c and e/f).

**3.3 $Q_{10}$ variability**

Seasonal variability was found in weekly $Q_{10}$ values, especially under the tree (Fig 5). In the alley, $Q_{10}$ ranged between 1.2 (warm months) and 2.0 (cold months), whereas under the tree, $Q_{10}$ ranged between 0.6 (warm months) and 1.8

(cold months). In general, the $Q_{10 \text{ Under Tree}}$ was higher and more variable than in the alley. Values for the entire study period were $Q_{10 \text{ Alley}} = 1.69 \pm 0.40$ and $Q_{10 \text{ Under Tree}} = 1.10 \pm 0.66$ (Mann Whitney test; $p < 0.01$). Hysteresis behavior was identified for $R_{s \text{ Alley}}$ in summer data when plotted with both $T_s$ and SWC. A linear relationship was found between the $Q_{10}$ and the SWC and soil temperature under the tree ($Q_{10} = 1.120 - 0.026T_s + 2.292SWC$; $R^2 = 0.36$) such that as the temperature increased and the SWC decreased, $Q_{10} < 1$ values were obtained. This relationship was not observed in the corridors.

[Figure]

**Figure 5: Seasonal variation in $Q_{10}$ parameter in the alley and under-tree chambers. Dashed lines are the weekly $Q_{10}$ values, and solid lines refer to the moving average daily values ($\pm$ 14 adjacent day window). The inner figure shows the relationship of weekly $Q_{10}$ to weekly average soil temperature and soil water content (SWC) for the two locations.**

**3.4 Rain Pulses events: $R_s$ and $R_{eco}$**

Of the seventy-five precipitation events, forty-one were accompanied by enhanced $R_s$. This was especially frequent when rain fell on dry soil, and the increased $R_s$ rates followed the longest inter-event periods (IEPs), as shown in Figure 6a. The PPT size does not have much influence on $\Delta R_s$ (data not shown). However, the four highest magnitude $R_s$ values were found in the lowest IEP and PPT in the alleys but were excluded from the regression as they were statistically classified as anomalies. In this way, notable rain pulses were detected when the IEP was large and SWC $< 15\%$ (Fig. 7a), whereas rainfall

[revised manuscript text omitted]

**3.6. Upscaled $R_{s,eco}$ vs. modeled $R_{eco}$.**

The response of net ecosystem exchange (NEE), soil respiration upscale to ecosystem level ($R_{s, eco}$), and ecosystem respiration ($R_{eco}$) to various rainfall pulses is represented in Figure 9 and we can observe that they respond in separate ways. The occurrence of a PPT event in September implied an increase in 5 µmol $CO_2$ m-2 s-1 of $R_{s,eco}$ and 3 µmol CO2  $m^{-2}$ $s^{-1}$ of NEE, whereas $R_{eco}$ did not respond (Fig. 9). Similar patterns occurred in the other rainfall events in November and June. These increases in $R_{s,eco}$ that were not reflected in $R_{eco}$ may  indicate an underestimation of respiration balances, as well as errors in $GPP_{eco}$. Regardless of rainfall, $R_{eco}$ was underestimated concerning $R_{s,eco}$ in the warm months, whereas in the cold months, the magnitudes of both approaches were similar (Fig. 10b and Fig. 10c). On average, $R_{s\ Under-Tree}$ represented 39% of $R_{s,eco}$ although the fraction was variable, with maximal contributions of 55% (winter) and minimal contributions of 25% (Fig. 10a). The percentage of "under tree" soil was 27%, which implies that in winter, half of the respiration of the olive grove was due to the soil that occupies a quarter of the ground.

[Figure]

**Figure 9: Response of CO₂ fluxes (daily average) to a precipitation event at three different times in the time series. Net ecosystem exchange (NEE), soil respiration upscaled to the ecosystem level (Rs, eco), night-time modeled ecosystem respiration (Reco - NT) and day-time modeled ecosystem respiration (Reco - DT).**

We show two approaches with the annually integrated $R_{eco}$ as the combination of measurement and empirical modeling based on EC data ($R_{eco-NT}$ =1310 ± 160 g C m$^{-2}$; $R_{eco-DT}$ = 850 ± 140 g C m$^{-2}$) and upscaling through chamber data ($R_{s,eco-}$ = 2100 ± 50 g C m$^{-2}$). During the warm months, the magnitudes of the chamber and night-time EC approaches are quite different, although they are consistent in the temporal variation. However, although the magnitudes are closer during the cold months, there is an inverse relationship between the two approaches. The daytime approach neither covaries with nor has similar magnitudes to respiration data from chambers in hot months. On the contrary, the response of $R_{eco-DT}$ is more like chambers the higher the influence of $R_{s-Under-Tree}$ on the ecosystem (Fig. 10a and Fig. 10c).

[Figure]

360 **Figure 10: a) Fraction of daily R$_{s\,Alley}$ and R$_{s\,Under-Tree}$ overestimated R$_{s,eco}$. b) Daily values of R$_{s,eco}$ and R$_{eco-NT}$ during the year (left) and the cumulative value of both (right). C) Daily values of R$_{s,eco}$ and R$_{eco-DT}$ during the year (left) and the cumulative value of both (right). The blue rectangle marks the warm period and high R$_s$ alley influence. The green rectangle is the cold period and high R$_{s\,Under\,Tree}$ influence.**

**4. Discussion**

365    This study relies on a dataset spanning one continuous and complete year of respiration fluxes at soil and ecosystem scales and provides significant insights into the temporal and spatial variations of olive grove respiration as well as influencing factors. A similar number of samples as the eddy covariance technique (~18,000 values per chamber per year) is much larger compared to the weekly or monthly measurement of typical studies, meaning that statistics such as the mean are much more rigorous. The abundance and continuity of half-hour measurements under trees and in alleys allow us to describe processes

370    and trends that have not been described in olive groves by typical studies based on manual campaigns.

**4.1 Spatial differences**

     Our findings affirmed a clear seasonal variability of R$_s$ and its main drivers (SWC, T$_s$), which is reflected in a high range of values compared with other studies using chambers. Bertolla et al. (2014) and Testi et al. (2008) measured daily values between 1.3 – 8.8 µmol m$^2$ s$^{-1}$ (n = 16) and 2.3 – 5.9 µmol m$^2$ s$^{-1}$ (n = 5; monthly) near the trunk of irrigated olive trees,

375    whereas our study showed a wider range of 3.2 - 23.9 µmol m$^2$ s$^{-1}$ (n = 365; daily). Such differences could be explained by the difference in the tree age (85 years for our individuals, versus primarily juvenile individuals between 2 and 7 years) which mean bigger and larger root systems. Juveniles will have less root development whose autotrophic and heterotrophic respiration is expected to be lower than that of an adult individual. In our study, we can deduce a predominant influence of respiration associated with the roots on the total soil activity, since R$_s$ was increasing as the measurements approached the trunk and R$_{s-UnderTree}$

380    exceeded on average three times that observed in the alleys on average. This excess changed during the year, being between 2 and 15 times higher during the warmest and the coldest months, respectively (Fig. S3a). Since the soil water content was similar in the cold period (Fig. S3c), the big differences could be due to i) heterotrophic respiration decreases in the alleys due to the additive effect of a higher Q$_{10}$ (Fig. 5) and a higher decrease in the temperature versus under the canopy (Fig. S3b); and ii) higher heterotrophic R$_{s-Under-Tree}$ due to a higher temperature compared to the alleys and differences in the substrate due

385    to the addition of root exudates and superficial leaf litter (Davidson and Janssens, 2006) meaning higher soil organic carbon under the tree canopy. The contribution of heterotrophic respiration to total respiration is complex to estimate (Comeau et al., 2018) and these data are not available. Nevertheless, considering that the magnitude of R$_{s-Alley}$ during the winter was very small, we assumed that the contribution of heterotrophic respiration under the tree is also small; therefore, the R$_{s\,Under-Tree}$ was largely controlled by rhizosphere respiration (R$_z$), which is the sum of heterotrophic respiration linked to the root system and

390    autotrophic respiration for maintenance and growth of the roots.

Continuous measurements allowed us to study the contribution of each location to the total $R_s$. Despite the total canopy fraction "Under Tree" being only 27% in our agroecosystem, there are periods where the proportion of $R_{s\ Under-Tree}$ contributes more than 50% to the $R_{s,eco}$ (Fig. 10a). Therefore, it cannot be assumed that the $R_{s\ Alleys}$ are representative of olive groves and most likely of mosaic tree crops such as savannas or 'dehesas'. In fact, in other systems with an open area/canopy distribution (Tang et al., 2005), $R_{s\ Under-Tree}$ was an order of magnitude greater than $R_{s\ -\ Alley}$; however, because the spatial gradients of $R_s$ were not quantified, the estimate at the ecosystem scale is unknown. In this study, we quantified the gradient between the measurements taken under the tree and in the alleys and found an exponential decrease as we moved away from the trunk that allowed us to do a simple upscaling. Although the influence of the roots extends gradually throughout the crop, its effect on respiration appears to be reduced significantly around a 3-m separation. Therefore, for this experimental site, the canopy radius (2.8m on average) can be a good proxy for determining the significant separation between the rhizosphere and alley. Other studies have established a random collar sampling map (for instance see Turrini et al., 2017) with different separation distances around the olive tree, which makes it difficult to integrate the role of rhizosphere respiration if distance correction factors are not applied (Supplementary material). Moreover, we found more respiration on the south side of the trees, where the temperature was also higher. Although the campaigns in which the gradient data were taken, only covered 4 of the 12 months of the year, and spatial differences may also vary over time, we have used the data obtained to weigh and scale the values of $R_s$ at the ecosystem scale and thus estimate $R_{s,eco}$.

**4.2 Eddy covariance models comparison**

The use of automatic chambers made it possible to assess annual balances of $R_s$. In the estimation of model $R_{eco}$ with data derived from EC, we obtain an annual balance of 850 gC m$^{-2}$ (Daytime approach) and 1300 gC m$^{-2}$ (Nighttime approach), whereas if we project $R_s$ at the ecosystem scale ($R_{s,eco}$), we obtain 2100 g C m$^{-2}$ (Fig. 12 b). The values obtained here with chambers may be similar to those found in grassland meadows (1990 g C m$^{-2}$; Bahn et al., 2008 ) and higher than other previous estimations in olive groves (860 ± 150 g C m$^{-2}$) although they were measured in monthly or bimonthly campaigns (Jian et al., 2021). A priori, $R_{s,eco}$ should be less than $R_{eco}$ because $R_{s,eco}$ is a fraction of $R_{eco}$ ($R_{eco} = R_{s,eco} + R_{AboveGround}$). However, the accumulated values obtained from the chamber ($R_{s,eco}$) are higher those obtained from the EC ($R_{eco}$), especially in summer. These differences between approaches could be related to the temporal mismatch between the two approaches. The chamber method takes snapshots of $R_s$, whereas the partitioning NEE method use a seven-day sliding window for their calculations. Furthermore, errors may be made in the NEE partitioning method, such as the underestimation of nocturnal fluxes due to low turbulence or the erroneous assumption that nocturnal respiration can be perfectly extrapolated to daytime respiration. On the other hand, during the day, we observe inverse relationships of $R_{s\ -\ Under\ Tree}/T_s$ to those that models based on NEE partitioning usually assume ($Q_{10}$ values < 1 in Fig. 5) and that could lead to an erroneous estimate of $R_{eco}$ by the daytime method. Moreover, the greater the role of above-ground tree respiration on $R_{eco}$, the worse the relationship between the $R_{s,eco}$, and $R_{eco\ -\ NT}$ (Fig. 10a and Fig. 10b), which indicates that the widely accepted partitioning model based on Reichstein et al. (2005) does not apply correctly to this semi-arid ecosystem. It is expected that Lasslop et al., (2010) would have greater agreement when the influence

of the $R_s$ Under Tree is greater (Fig. 10) since $R_{eco}$ calculates it from the photosynthetic organisms' activity. Since each model is based on different mechanisms, we can say that no method faithfully represents the ecosystem, and furthermore there are more drivers than temperature and SWC and these are also interrelated. Therefore, more research in the application of these models in others semi-arid systems is necessary. Despite that, when a multi-chamber system is not available, we recommend for this type of ecosystem the use of daytime models in cold seasons and the use of nighttime models in hot seasons when most of the contribution to $R_s$ comes from the heterotrophic respiration of the alleys.

**4.3 $R_s$ drivers**

$Q_{10}$ differs in the alley and tree-base in terms of magnitude and seasonal evolution (Fig. 5). In the alley $Q_{10} > 1$ was always found, but under the canopy, we found periods with $Q_{10}$ values close to 1. This means that the variation in $R_s$ during this period is decoupled from changes in soil temperature. In addition, $Q_{10} < 1$ values during summer indicated that respiration decreased as temperature increased. That is, the respiration of the rhizosphere of the tree canopy was associated with soil temperature. Therefore, the traditional parameter $Q_{10}$, determined through field measurements of $R_s$ and temperature (Davidson and Janssens, 2006) cannot be used to define the respiration of Mediterranean ecosystems because of their large spatial and temporal heterogeneity where there are plants that inhibit their respiration at high temperatures. The $Q_{10}$ values obtained in this study were calculated using 7-day windows and soil temperature measurements at a depth of 5 cm. However, these results may vary depending on the length of the time window and the depth of the temperature sensors, as the temperature propagation through the soil introduces a time lag that can influence the estimates (Barron-Gafford et al., 2011; Hamerlynck et al., 2013).. It is currently known, variations in $Q_{10}$ are controlled by soil and vegetation factors and not only climate (Chen et al., 2020). Therefore, although the global value of $Q_{10}$ is estimated to be 1.5 (Bond-Lamberty and Thomson, 2010), the variability of reported $Q_{10}$ varies enormously, reaching values higher than 200 in ecosystems with very low temperatures (Mikan et al., 2002). Conversely, $Q_{10}$ values less than 1 have been found as well in other regions with semi-arid climates that include a dry period such as continental monsoon (Han and Jin, 2018), suggesting that it is not exclusively found in olive groves, but may be common in water-limited ecosystems with dry periods. In addition, the regressions used to obtain $Q_{10}$ are usually not good when the water content in the soil is low (Wang et al., 2014) and the $R_s$-$T_{soil}$ relationship disappears or even becomes negative as we can see in figure 5.

The areas with the greatest uncertainty in global $R_s$ prediction models are semi-arid regions (Warner et al., 2019), where water acts as a limiting factor and $R_s$ decreases even with increasing temperature (Zhao et al., 2017). For example, in Mediterranean mountain grasslands, temperature is no longer a good predictor (Bahn et al., 2008). In our study, we observed a coincidence in the alley of the reduction of $R_s$ with the prolonged decrease in soil moisture in the prolonged summer drought (July - September), indicating a connection between $R_s$ and humidity (Fig. 3). Thus, even if the temperature increases in July, $R_s$ in the two compartments appears to decrease. However, $R_{s – Under-Tree}$ also decreased with the advance of summer even though $SWC_{Under-Tree}$ remained relatively constant due to irrigation. Also, $R_{s – Under-Tree}$ was practically constant throughout the day

except in summer when there is a negative relationship with temperature. This decoupling can be explained by a reduction of root exudates due to reduced photosynthesis of olive trees that could be induced by an increase in VPD. It is known that tree photosynthesis modulates soil respiration (Högberg et al., 2001) as metabolic activity is related to the closure of stomata at high temperatures (Tang et al., 2005; Makita et al., 2018). $CO_2$ assimilation decreases with high VPD values when olive trees close their stomata (Fernández and Moreno, 1999), affecting the NEE (Chamizo et al., 2017). Stomatal closure has an impact on the rhizosphere because it inhibits the transport of photosynthetic products or carbohydrates from photosynthesis (which in turn depends on ecophysiological and meteorological factors) by the phloem, decreasing root activity and exudates, thus decreasing rhizosphere respiration ($R_z$). Therefore, $R_z$ may be dynamically linked to vegetative growth, climate, or competition, all linked to $CO_2$ assimilation capacity of olive trees (Aranda-Barranco et al., 2023). Tang et al. (2005) established that the translocation time of photosynthetic products from leaves to roots can be between 7 and 12 h. The lag for the isotopic signal of photosynthesis in trees to appear at $R_s$ is in the range of days in other ecosystems (Ekblad and Högberg, 2001), which seems to indicate that stomatal closure would have an immediate effect (hours) on the reduction in $CO_2$ transport to the rhizosphere, whereas it would have a later effect (days) on the reduction of root exudates. The reduction in $R_s$ at high VPD values is observed both at night and during the day and at times with and without irrigation. Nevertheless, the association between photosynthesis and $R_{s - Under-Tree}$ can be confused with the relationship between $T_s$ and $R_s$ and more study is needed in this regard to establish the connection between VPD and $R_z$.

Therefore, the response of $R_s$ to temperature fluctuations is influenced not only by soil temperature, making it crucial to consider additional factors such as SWC, photosynthesis, or precipitation events when modeling $R_s$ in Mediterranean environments (González-Ubierna and Lai, 2019). In our study, we see that the variations of SWC are higher in the alleys (Fig. S2a); therefore, the variability of the relationship between $R_s$ and $T_s$ will be higher if they are conditioned by humidity. In addition, the drivers are interrelated as temperature-dependent responses that are further influenced by soil moisture and precipitation (Hursh et al., 2017). Furthermore, we identified hysteresis behavior only in summer in alleys, which could indicate diurnal changes of SWC close to some critical value for the $R_s/T_{soil}$ relation. However, finding the hysteresis pattern also with SWC indicates that there is another factor in addition to temperature involved in $R_s$.

**4.4 Rain Pulse Events**

Although the main transport process is molecular diffusion, rain pulse events can also produce an immediate release of $CO_2$ by displacing gas within the pores (Inglima et al., 2009). Marañón-Jiménez et al. (2011) suggest that a significant portion of the $CO_2$ released within the first two hours after water is added likely originates from the degassing of CO2-rich air trapped in the soil pores. $CO_2$ adsorbed to the surface of soil particles (Ravikovitch et al., 2005) and stimulation of microbial activity can increasing soil respiration (Jarvis et al., 2007). Rain pulses can be the most important driver in terms of the seasonal trend of soil $CO_2$ efflux in semi-arid ecosystems (Leon et al., 2014). We can see that water is a limiting factor for $R_s$ with a large rainfall pulse after the summer drought (first rectangle on the left; Fig. 3); however, in periods with high SWC, this

relationship is lost (Fig. 7). Other studies in olive grove alleys (Testi et al., 2008; Sierra et al., 2016; Chamizo et al., 2017) have shown $R_s$ values between 0.5-1.6 µmol m$^2$ s$^{-1}$ ($n_{collar}$– ~ 10) based on field campaign measurements taken outside of rainy days. In contrast, using automatic measurements, we found a higher variability with 0.4 - 11.3 µmol m$^2$ s$^{-1}$ values (n = 365), but with a median of 1.5 µmol m$^2$ s$^{-1}$ (Fig. 2a), within the range of the other studies. This reflects the fact that continuous measurements can detect rain pulse events that tend to fall outside the usual ranges. The conventional daytime and nighttime methods for $R_{eco}$ partitioning via Eddy covariance data modeling appear not to respond to rainfall pulses, whether the soil was previously dry or not (Fig. 8d), and this becomes evident when we observe how only the chamber fluxes respond to a PPT event.

 In our study, pulses of rain were detected on 11% of the days of the year, which implied that up to 18% of $CO_2$ emissions occurred on days with pulses of rain which implied that 15% of $R_{s,eco}$ emissions came from rain pulses. However, the variability in pulse length was high, with pulses lasting between the high intensity moment of the first hour and up to several days. Given that the duration of the pulses is usually between 3 and 6 h at our site and the intensity of rain pulse events decreases with successive events, we can be more cautious and estimate that the total ecosystem contribution of rain pulses is less than 15%, as is the case of others semi-arid areas ( Delgado-Balbuena et al., 2023) where 5% contribution was estimated. At the ecosystem scale, we observed slight pulse signals with a lag of several days; therefore, the NEE models based exclusively on radiation or temperature may not be the most accurate for real-time characterization of this phenomenon. However, incorporating soil water content into these models would significantly enhance their predictive ability for the Birch effect.

[revised manuscript text omitted]

---

## Author Response (AR3)

Please see below the response to the reviewer regarding their concerns. We also included a PDF version of the revised manuscript with changes highlighted in yellow.

**Comments to the author:**

**Dear authors, thank you for addressing all of the comments from the reviewers. Please check a few more minor corrections to wording below, and we can proceed with accepting the paper once corrected.**

**L159 — Needs slight rewording, for example, "only relative values in the Alley/…"**

    R: Done

**L306 — Change to "the increase occurred when SWC < 20% and was less intense". Also consider changing "less intense" to "less steep" or "less pronounced".**

    R: Done

**L278, Fig. 4 — This caption could still be clarified further. As a second sentence, please add "Each panel represents a tree-alley pair" as mentioned in your response to the reviewer. The total number of chambers could also be mentioned in the caption (as on L269). Also, consider rewording the 1st sentence as "…chambers over one week in July (a,b,c) and one week in April (d,e,f)."**

    R: Done

**L510 — I would suggest changing "less noticeable" to "not statistically significant". In general, here and elsewhere "noticeable" isn't a very informative or statistically rigorous term.**

    R: Done

**L538 — Change "a reduce in" to "a reduction in" at a minimum, but also consider rewording this sentence for clarity. For example: "Although the underlying processes driving this observation need further study, …"**

    R: Done